# F2M-Reg: Unsupervised RGB-D registration with Frame-to-Model Optimization

## Abstract

This paper focuses on training a robust RGB-D registration model without ground-truth pose supervision. Existing methods usually adopt a pairwise training strategy based on differentiable rendering, which enforces the photometric and the geometric consistency between the two registered frames as supervision. However, this frame-to-frame framework suffers from poor multi-view consistency due to factors such as lighting changes, geometry occlusion and reflective materials. In this paper, we present F2M-Reg, a novel frame-to-model optimization framework for unsupervised RGB-D registration. Instead of frame-to-frame consistency, we leverage the neural implicit field as a global model of the scene and use the consistency between the input and the rerendered frames for pose optimization. This design can significantly improve the robustness in scenarios with poor multi-view consistency and provides better learning signal for the registration model. Furthermore, to facilitate the neural field optimization, we create a synthetic dataset, Sim-RGBD, through a photo-realistic simulator to warm up the registration model. By first training the registration model on Sim-RGBD and later unsupervisedly fine-tuning on real data, our framework enables distilling the capability of feature extraction and registration from simulation to reality. Our method outperforms the state-of-the-art counterparts on two popular indoor RGB-D datasets, ScanNet and 3DMatch. Code and models will be released for paper reproduction.

## 1 Introduction

The difficulty of 3D data acquisition has significantly diminished owing to the substantial increase in RGB-D sensor availability and a concurrent decrease in costs. The prolific collection of RGB-D data has greatly propelled the advancement of deep learning in the field of 3D vision, resulting in substantial improvements in the performance of applications such as RGB-D SLAM and RGB-D reconstruction. A pivotal challenge in achieving reliable 3D reconstruction based on discrete RGB-D image frames lies in establishing correct inter-frame associations, through means such as feature matching, to facilitate camera pose estimation. Motivated by these, our goal is to devise a robust registration model for RGB-D images, thereby providing robust and high-quality pixel-level matching for RGB-D registration.

Traditional methods, relying on hand-crafted features (such as SIFT (Lowe, 2004)) find difficulty in handling complex and noisy real-world data. Deep learning based methods, on the other hand, have gained much attention lately and most works adopt a supervised learning approach (Qin et al., 2022; Yu et al., 2021; 2023; Ao et al., 2021) to accomplish robust registration. These learning-based methods can support RGB-D registration between frames even with very small overlaps. The performance of supervised learning approaches, however, depends highly on the quality of the data annotation, i.e., ground-truth frame poses, which are difficult to obtain and hence limit their application in practice.

To overcome the reliance on annotated data in learning-based methods, the exploration of better strategies to extract information from unlabeled data for achieving unsupervised learning in RGB-D registration has gradually become a research focus. Inspired by works in multi-view geometry, studies have found that the geometric and photometric consistency inherent in the RGB-D sequences of a scene can offer effective supervision for feature extraction. To our knowledge, UR&R(El Banani et al., 2021) is the first work proposing an unsupervised framework for RGB-D point cloud registra-

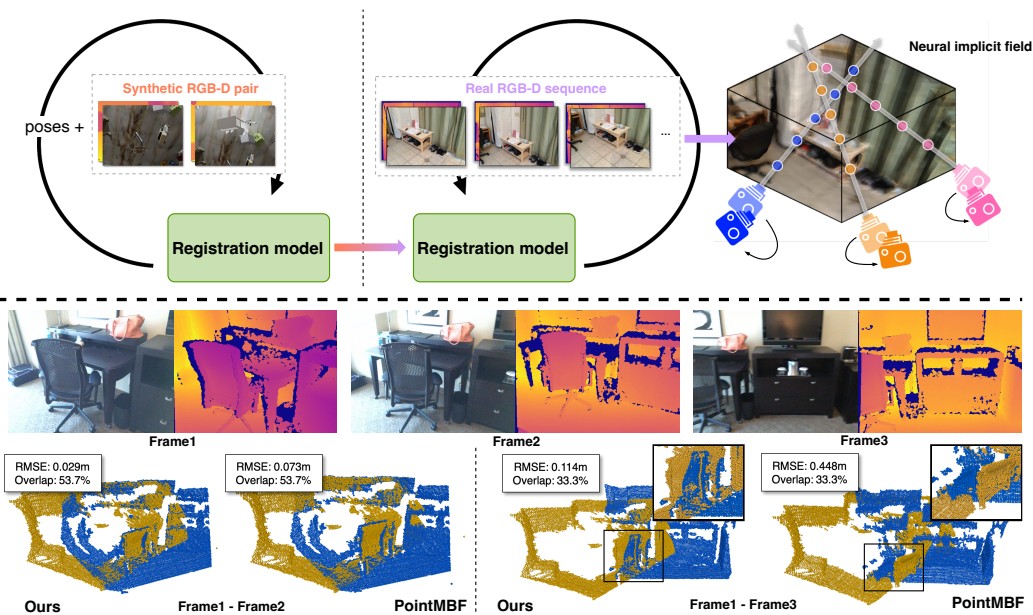

Figure 1: We propose F2M-Reg, a frame-to-model optimization framework for unsupervised RGB-D registration. The registration is first warmed up with synthetic data, and then fine-tuned on real-world data in the frame-to-model manner (top). Although the frame-to-frame method can successfully register the easy case (bottom-left), it cannot register the case with lighting changes and low overlap (bottom-right). On the contrary, our method effectively register the hard case.

tion. It takes two RGB-D frames with overlap as input and estimates their relative pose with a registration model (Wang et al., 2022; Yuan et al., 2023). Based on this relative pose, UR&R rerenders one frame to the reference frame of the other with a point cloud-based differentiable rasterization mechanism, and enforces the photometric and geometric consistency between the rerendered and the input frames to enable training of the registration model. However, this frame-to-frame optimization lacks global contextual information of the entire scene, especially in cases with limited distinctive features, making it vulnerable to scenarios with poor multi-view consistency due to factors such as lighting and occlusion (see Fig. 1). Furthermore, point cloud-based methods require small overlap to enable high-quality rerendering which limits its applicability in more challenging cases.

We introduce Unsupervised RGB-D **Reg**istration framework with **F**rame-to-**M**odel Optimization (**F2M-Reg**). To overcome the limitations of frame-to-frame optimization, we adopt the neural implicit field as the global model to support unsupervised training. Note, however, that the initialization of the neural field requires accurate frame poses but we cannot achieve this without a good registration model, which makes a *chicken-and-egg* problem. So we opt to utilize synthetic RGB-D data rendered with 3D scene models to train an initial registration model to warm up the frame-to-model optimization. To this end, we create a synthetic dataset, Sim-RGBD, with photo-realistic rendering of CAD models, which contains more than 100k rendered images of 90 scenes. As shown in Fig 1, the registration model is first trained on Sim-RGBD with the ground-truth poses and later unsupervisedly fine-tuned on the real-world data in a frame-to-model manner. As the neural field is constructed from the entire RGB-D sequence, it can better handle multi-view inconsistency factors such as lighting changes, geometry occlusion and reflective materials. Therefore, enforcing the photometric and geometric consistency between the rerendering and the input frames can better optimize the estimated poses than the frame-to-frame methods, which enhances the learning signal for the registration model. This refining stage enables distilling the capability of feature extraction and registration from simulation to real world.

We have evaluated our method on two popular indoor RGB-D datasets, ScanNet(Dai et al., 2017) and 3DMatch(Zeng et al., 2017). We demonstrate that our method outperforms both traditional and recent unsupervised learning-based registration pipelines. Moreover, our method achieves signifi-

cantly better performance than previous methods in more challenging scenarios with lower overlap or severe lighting changes. Extensive ablation studies are conducted to prove the effectiveness of different components of our pipeline. In summary, our contributions are as follows:

- We propose a frame-to-model optimization framework guided by a neural implicit field for unsupervised RGB-D registration. The infusion of global reconstruction information enhances the reliability of rerendering, which fortifies the robustness of registration model.
- We devise a synthetic warm-up mechanism to provide high-quality initial poses for neural implicit field optimization and create a synthetic dataset for warming up registration model.
- Our method achieves new state-of-the-art results on the two popular indoor RGB-D datasets, ScanNet and 3DMatch.

## 2 RELATED WORK

### 2.1 POINT CLOUD REGISTRATION

Point cloud registration is a problem of estimating the transformation matrix between two frames of scanned point clouds. The key lies in how to detect features with specificity from the two-frame point cloud. Since deep learning has been found good at feature representation, how to learn robust and invariant visual features through deep learning networks has become a focus of research. (Wang et al., 2020; Qi et al., 2017a;b; Duan et al., 2022; Pan et al., 2021; Qin et al., 2022) Many Feature learning methods (Wang et al., 2022; Deng et al., 2018; Gojcic et al., 2019; Ao et al., 2021; Bai et al., 2020; Choy et al., 2020; 2019; Qin et al., 2022; Yew & Lee, 2022; Yu et al., 2023) were proposed. They get the point cloud features by neural network and use a robust estimator e.g. RANSAC to estimate the rigid transformation. Different from focusing on feature learning, there are some end-to-end learning-based registration methods (Yang et al., 2019; Wang et al., 2019; Elbaz et al., 2017; Lu et al., 2019; Huang et al., 2020) that treat the registration as a regression problem. They encoded the transformations into the implicit space as a parameter in the network optimization process.

### 2.2 UNSUPERVISED POINT CLOUD REGISTRATION

The aforementioned methods rely on ground-truth poses to supervised the training. The ground-truth pose is often obtained by reconstruction of the SfM, which suffers from high computational overhead and instability. Recently, unsupervised RGB-D registration methods have been proposed to bypass the need of pose annotations. To our knowledge, UR&R(El Banani et al., 2021) is the first unsupervised registration framework by introducing a differentiable render-based loss to optimize the feature extractor. BYOC(El Banani & Johnson, 2021) stands for the fact that randomly initialized CNNs also provide relatively good correspondences, proposed a teacher-student framework to train their feature extractor. LLT(Wang et al., 2022) fused the geometric and visual information in a more trivial way by introducing a multi-scale local linear transformation to fuse RGB and depth modalities. PointMBF(Yuan et al., 2023) has designed a network based on unidirectional fusion to better extract and fuse features from geometric and visual sources and has achieved state-of-the-art performance. However, these methods have difficulty in handling multi-view inconsistency caused by factors such as lighting changes, highlight or occlusion.

### 2.3 POSE OPTIMIZATION IN NEURAL SLAM

Existing Neural SLAM methods (Sucar et al., 2021; Zhu et al., 2022; Tang et al., 2023; Wang et al., 2023; Zhang et al., 2023; Yang et al., 2022; Johari et al., 2023) incorporate neural implicit representations into RGB-D SLAM systems, allowing tracking and mapping from scratch. The groundbreaking work, iMAP (Sucar et al., 2021), encode both the color and geometry of the scene into a MLP. This MLP can be jointly optimized with a batch of poses through rendering loss. In the subsequent works, NICE-SLAM (Zhu et al., 2022) and Vox-Fusion (Yang et al., 2022) introduce a hybrid representation that combines learnable grid-based features with a neural decoder, enabling the utilization of local scene color and geometry to guide pose optimization. More recently, Mips-fusion (Tang et al., 2023) proposed a robust and scalable RGB-D reconstruction system with a multi-implicit-submap neural representation. Co-SLAM (Wang et al., 2023) proposed a joint coordinate and sparse-parametric encoding and a more global bundle adjustment approach. Inspired by

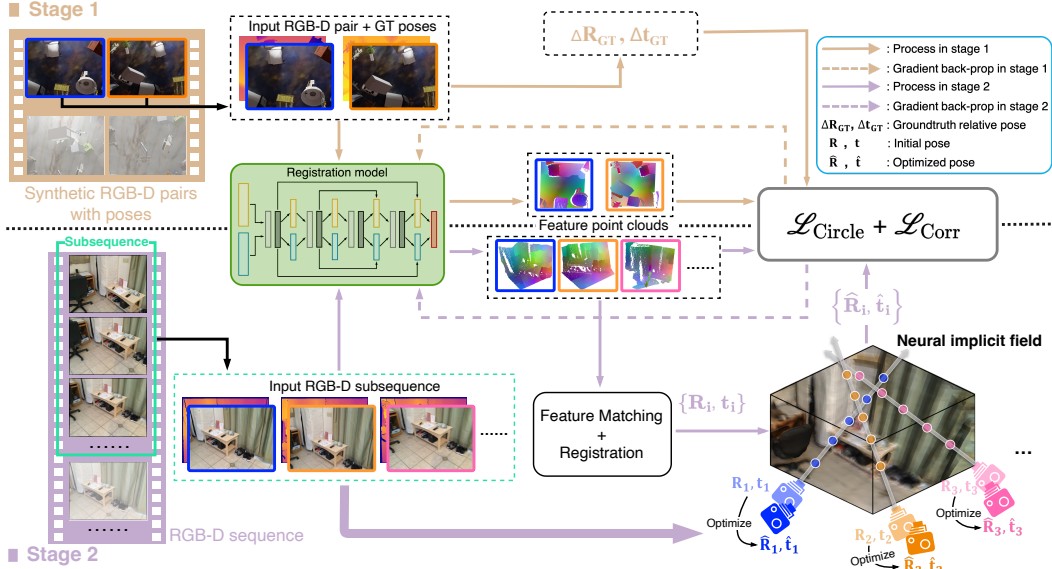

Figure 2: Overall pipeline of F2M-Reg. Our framework can be divided into two stages. The first synthetic warm-up stage leverages synthetic RGB-D pairs as well as their ground-truth poses to train the registration model in a supervised manner. In the second frame-to-model optimization stage, we take an RGB-D sequence as input and use the registration model to estimate the relative pose for every two consecutive frames. Based on the estimated poses, we jointly optimize a neural implicit field of the whole scene and the estimated poses. At last, the optimized poses are used to fine-tune the registration model on real-world data.

the aforementioned works, we introduce our framework for estimating the initial camera pose using a feature extractor and subsequently refining the pose through implicit 3D reconstruction.

## 3 METHOD

### 3.1 OVERVIEW

Given two RGB-D frames $\mathcal{X} = (\mathbf{I}^{\mathcal{X}}, \mathbf{X})$ and $\mathcal{Y} = (\mathbf{I}^{\mathcal{Y}}, \mathbf{Y})$, where $\mathbf{I}^{\mathcal{X}}, \mathbf{I}^{\mathcal{Y}} \in \mathbb{R}^{H \times W \times 3}$ are the RGB images and $\mathbf{X} \in \mathbb{R}^{N^{\mathcal{X}} \times 3}, \mathbf{Y} \in \mathbb{R}^{N^{\mathcal{Y}} \times 3}$ are the point clouds backprojected from the corresponding depth images, our goal is to recover the 6-DoF relative pose $\mathbf{T} \in \mathcal{SE}(3)$ between them, which consists of a 3D rotation $\mathbf{R} \in \mathcal{SO}(3)$ and a 3D translation $\mathbf{t} \in \mathbb{R}^3$. To solve this problem, a learning-based registration model $\mathcal{F}$ first extracts point features and retrieves point correspondences $\mathcal{C} = \{(\mathbf{p}_i, \mathbf{q}_i) \mid \mathbf{p}_i \in \mathbf{X}, \mathbf{q}_i \in \mathbf{Y}\}$ via feature matching. The relative pose is then estimated based on the correspondences. Obviously, the discriminativeness of the extracted features accounts for the quality of the resultant relative pose. However, the training of $\mathcal{F}$ heavily relies on the ground-truth pose $\mathbf{T}^* = \{\mathbf{R}^*, \mathbf{t}^*\}$, which suffers from great annotation difficulty and unstable convergence.

In this work, we propose an unsupervised point cloud registration method named *F2M-Reg*. Our method leverages unposed RGB-D sequences to train the registration model $\mathcal{F}$. We first describe our registration model in Sec. 3.2. To achieve effective supervision, we generates high-quality relative pose in a frame-to-model manner (Sec. 3.3). To inspire the parameters of $\mathcal{F}$, we build a scene-level synthetic dataset and pretrain $\mathcal{F}$ on this dataset so that reasonable initial features can be learned (Sec. 3.4). Fig. 2 illustrates the overall pipeline of our method.

### 3.2 REGISTRATION MODEL

Our registration model $\mathcal{F}$ adopts a two-branch feature encoder, *i.e.*, the visual branch and the geometric branch, which fuses the information from both the visual (2D) and the geometric (3D) spaces for better feature distinctiveness, similar to PointMBF (Yuan et al., 2023). The visual branch is

a modified ResNet-18 (He et al., 2016) network, following a U-shape architecture. The geometric branch is a KPFCN (Bai et al., 2020; Thomas et al., 2019) network symmetric with the visual branch. Both branches adopt a three-stage architecture, and a PointNet-based fusion module fuses the features from the two modalities after each stage. The point features of the two frames are, respectively, denoted as $\mathbf{F}^{\mathcal{X}} \in \mathbb{R}^{N^{\mathcal{X}} \times C}$ and $\mathbf{F}^{\mathcal{Y}} \in \mathbb{R}^{N^{\mathcal{Y}} \times C}$, which are $\ell_2$-normalized onto a unit hypersphere. We then extracts the correspondences by mutual matching based on the point features, and select the top-$N_C$ correspondences with the smallest distances in the feature space, which are fed into RANSAC to estimate the final relative pose.

## 3.3 Unsupervised Registration with Frame-to-Model Optimization

An unsupervised registration pipeline usually first gives a rough estimation of the pose, and then supervise the registration model with the estimated pose. Obviously, the quality of the estimated pose significantly affects the accuracy of the trained registration model. Existing methods (El Banani et al., 2021; Wang et al., 2022; Yuan et al., 2023) use differentiable rasterization and optimize the frame pose according to the photometric and the geometric consistency between two nearby frames from an RGB-D sequence. Nevertheless, the consistency between two frames are easily to be affected by occlusion or lighting changes under different viewpoints, which fails to effectively refine the frame poses and thus harms the training of the registration model. This has inspired us that a more comprehensive modeling of the whole scene is required to effectively optimize the frame poses, *i.e.*, *frame-to-model* optimization. Recently, neural implicit field (Mildenhall et al., 2021; Wang et al., 2021) has the ability to model the lighting and geometric structures in a scene, and jointly optimize 3D maps and poses (Zhu et al., 2022; Wang et al., 2023). Based on this insight, we propose to train the registration model scene by scene and optimize a neural implicit field as a global model for each scene for better pose refinement. This allows to optimize the poses in a frame-to-model fashion instead of the traditional frame-to-frame one, which can better handle the occlusion and lighting changes.

**Training pipeline.** As shown in Fig. 2, to avoid the error accumulation and the huge time overhead caused by joint map-pose optimization in large scenes, we opt to process small subscenes instead of the whole scene. Specifically, we split the RGB-D sequence of a scene into subsequences of 200 frames, and we optimize a neural implicit field $\mathcal{M}$ for each subsequence. Within each subsequence, we further sample keyframes every 20 frames for training and all other frames are omitted. The reference frame of the first keyframe is treated as the global reference frame of the subscene. For each keyframe, we first register it with the previous keyframe with $\mathcal{F}$ to obtain its initial pose, and then insert it into $\mathcal{M}$ to jointly optimize its pose and the map. At last, we use the optimized pose of each keyframe to supervise the registration model.

**Initial pose generation.** Given the point features $\mathbf{F}^{\mathcal{X}}$ and $\mathbf{F}^{\mathcal{Y}}$, for each point $\mathbf{x}_i \in \mathbf{X}$, we then find its nearest point $\mathbf{y}_{n_i} \in \mathbf{Y}$ in the feature space as a correspondence. The weight for each correspondence is computed as:

$$w_i = 1 - \frac{\|\mathbf{f}_i^{\mathcal{X}} - \mathbf{f}_{n_i}^{\mathcal{Y}}\|}{2}. \tag{1}$$

At last, we select the top $k$ correspondences with the largest weights. The same computation goes for $\mathbf{Y}$. As a result, we obtain $2k$ correspondences, denoted as $\mathcal{C}$. To compute the initial pose, we randomly sample 10 correspondence subsets of 20% of the correspondences. For each subset, we use weighted SVD (Besl & McKay, 1992) to compute a pose hypothesis and select the best pose which minimizes:

$$E = \sum_{(\mathbf{p}_i, \mathbf{q}_i) \in \mathcal{C}} w_i \|\mathbf{R}\mathbf{p}_i + \mathbf{t} - \mathbf{q}_i\|. \tag{2}$$

**Pose optimization.** We adopt a neural implicit field similar with Co-SLAM (Wang et al., 2023) due to its advances in the speed and the quality of reconstruction. Our neural field maps the world coordinates $\mathbf{x} = (x, y, z)$ and the viewing direction $\mathbf{d} = (\theta, \phi)$ into the color $\mathbf{c}$ and the TSDF value $s$. Following the SLAM pipeline, our method is split into the *Tracking* stage and *Mapping* stage.

In the tracking stage, we optimize the pose of the keyframe with the neural implicit field $\mathcal{M}$. The optimized pose in this stage is named the *tracked pose*, denoted as $\tilde{\mathbf{T}}_i$. For the $i$-th keyframe, we first calculate its untracked pose $\mathbf{T}_i = \Delta\mathbf{T}_{i-1,i} \cdot \hat{\mathbf{T}}_{i-1}$, where $\hat{\mathbf{T}}_{i-1}$ is the mapped pose of the previous

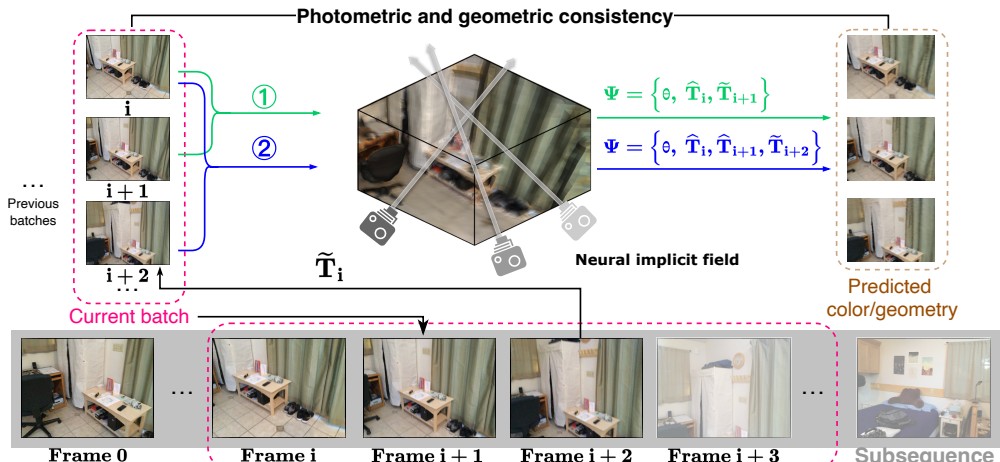

Figure 3: Mapping stage. The first frame in the current batch, Frame $i$, can either be the first of a new subsequence or the last from the previous batch, with its pose known. Once a new frame is tracked, it is added to the batch with its tracked pose $\tilde{\mathbf{T}}_i$. In Step 1, when the $(i+1)^{\text{th}}$ frame is added, its tracked pose $\tilde{\mathbf{T}}_{i+1}$ is optimized along with the mapped pose $\hat{\mathbf{T}}_i$ and the implicit scene representation $\theta$. In Step 2, after adding the $(i+2)^{\text{th}}$ frame, the optimization parameters become $\mathbf{\Psi} = \{\theta, \hat{\mathbf{T}}_i, \hat{\mathbf{T}}_{i+1}, \tilde{\mathbf{T}}_{i+2}\}$, with further frames tracked and optimized similarly.

keyframe as described later and $\Delta\mathbf{T}_{i-1,i}$ is their initial relative pose from the registration model. $\mathbf{T}_i$ is then optimized to $\tilde{\mathbf{T}}_i$ by supervising the photometric and the geometric consistency between the input RGB-D frame and the rerendered frame by the $\mathcal{M}$. The neural implicit field $\mathcal{M}$ are fixed in this stage. Please refer to the supplementary material for more details about the neural implicit field $\mathcal{M}$ training. As the filed implicitly models the whole scene, this frame-to-model paradigm could alleviate the influence of heavy occlusion or lighting changes from different viewpoints, and thus achieves more effective optimization of the keyframe pose.

After one keyframe is tracked, we jointly optimize the $\mathcal{M}$ and the poses of the keyframes in the mapping stage. The pose refined in this stage is named the *mapped pose*, denoted as $\hat{\mathbf{T}}_i$. The mapping stage adopts a batch-wise optimization strategy. When a keyframe is tracked, it is added into the current batch. Then all keyframes in this batch are used to optimize the $\mathcal{M}$ to improve the implicit scene model, with their poses being optimized simultaneously. This joint optimization further improves the quality of the keyframe poses. After we have collected a maximal batch size $B$ of keyframes, we train the registration model with the mapped poses of the keyframes in the current batch. The batch is then emptied except the last keyframe, which is used to provides the anchor pose for the coming keyframes in the next batch.

**Training the registration model.** After obtaining the optimized poses of a batch, we compute the relative poses between consecutive keyframes which are used as the frame poses to train the registration model. We first compute correspondences between two keyframes with their optimized poses and then apply the circle loss (Sun et al., 2020; Huang et al., 2021) and the correspondence loss (Eq. 2) during training. Please refer to the supplementary material for more details.

### 3.4 SYNTHETIC WARM-UP

Jointly optimizing the neural implicit field $\mathcal{M}$ and the poses of keyframes requires relatively accurate initial poses. However, a randomly initialized registration model tends to generate enormous outlier correspondences. This causes the initial poses to be erroneous, and thus leads to suboptimal convergence. To address this issue, we propose to leverage synthetic data to warm up the registration model. With the synthetic data, we can warm up the registration model in supervision by the ground-truth poses so that it can provide reasonable initial poses.

**Sim-RGBD dataset.** To warm up the model training, we first construct a synthetic dataset using photo-realistic simulation with BlenderProc (Denninger et al., 2023), named *Sim-RGBD*. Sim-RGBD consists of 90 scenes, which are split into 60 training scenes and 30 validation scenes. Specifically, for each scene, we create two boxes centered at $(0, 0, 0)$ in the sizes of respectively $10\text{m} \times 10\text{m} \times 5\text{m}$ and $6\text{m} \times 6\text{m} \times 3\text{m}$, and uniformly select 400 positions in the space between them. We then place a random object model from ShapeNet dataset (Chang et al., 2015) at each position, which is randomly rotated, translated, and scaled.

After constructing the synthetic scenes, we render 400 pairs of RGB-D frames from each scene. Another problem here is how to sample appropriate camera poses to ensure the synthetic pairs are more realistic. To this end, we opt to first sample the pose of the source frame, and then sample the relative pose between the source and the target frames. For the source pose, the camera direction is determined by a random pitch angle between $[15°, 75°]$ and a random yaw angle between $[0°, 360°]$. And the camera position is determined by a random distance between $[0.7\text{m}, 1.5\text{m}]$ from $(0, 0, 0)$ along this direction. For the relative pose, we first randomly sample a rotation axis, and then sample the rotation angle from $\mathcal{N}(20°, 15°)$ and the translation from $\mathcal{N}(0.4\text{m}, 0.2\text{m})$. As the two rendered frames could have little overlap, we only preserve the pairs with the overlap ratio above 0.3. As shown in Sec 4.3, the synthetic scenes simulated with this simplistic strategy effectively warm up the model.

**Training settings.** Similar to Sec. 3.3, we use the ground-truth poses to retrieve correspondences and apply the circle loss to train the registration model. Note that it is important to ensure the warming up stage and the frame-to-model optimization stage to use the same training strategy. Otherwise, the two models could be in different feature spaces, thus harming the final performance.

## 4 EXPERIMENTS

We conduct our experiments on three RGB-D datasets: 3DMatch (Zeng et al., 2017), ScanNet (Dai et al., 2017), and Sim-RGBD, as mentioned above. The experimental content is structured as follows. First, we present the experimental design in Sec. 4.1. Next, we outline the experimental outcomes in Sec.4.2, demonstrating the superiority of our method with supporting results. Following this, we perform comprehensive ablation studies in Sec.4.3 to analyze the specific contributions of each module in our design. Finally, qualitative results are presented in Sec.4.4.

### 4.1 EXPERIMENTAL SETTINGS

**Datasets.** We conduct extensive experiments on two large real-world datasets, 3DMatch (Zeng et al., 2017) and ScanNet (Dai et al., 2017), as well as a synthetic dataset, Sim-RGBD. For the ScanNet dataset (Dai et al., 2017), we follow its original training/testing/validation split, dividing it into 1045/312/100 scenes, respectively. Similarly, for the 3DMatch dataset (Zeng et al., 2017), we adhere to its original split, dividing it into 71/19/11 scenes for training, testing, and validation. The Sim-RGBD dataset is split into 60 scenes for training and 30 scenes for testing.

**Metrics.** We follow previous work (Yuan et al., 2023; Wang et al., 2022; El Banani et al., 2021) and use rotation error, translation error, and chamfer error as our evaluation metrics. Each metric is reported with three different thresholds, along with both mean and median values. Additionally, we adopt the settings from (El Banani et al., 2021; Wang et al., 2022; Yuan et al., 2023) to generate view pairs by sampling image pairs that are 20 frames apart. We also evaluate view pairs sampled 50 frames apart.

**Baseline Methods.** We compare our method with baselines from three categories, (1) traditional methods: ICP (Besl & McKay, 1992), FPFH (Rusu et al., 2009) SIFT (Lowe, 2004), (2) learning-based supervised methods: SuperPoint (DeTone et al., 2018), FCGF(Choy et al., 2019), DGR (Choy et al., 2020), 3D MV Reg (Gojcic et al., 2020) and REGTR (Yew & Lee, 2022), and (3) unsupervised methods: UR&R (El Banani et al., 2021), BYOC (El Banani & Johnson, 2021), LLT (Wang et al., 2022) and PointMBF (Yuan et al., 2023). We use the results of the baselines reported by (Wang et al., 2022; Yuan et al., 2023).

Table 1: **Pairwise registration on ScanNet Dai et al. (2017) dataset with a 20 frames apart**. Sup means supervision.

| | Train Set | Sup | Rotation(°) | | | | | Translation(cm) | | | | | Chamfer(mm) | | | | |
|---|---|---|---|---|---|---|---|---|---|---|---|---|---|---|---|---|---|
| | | | Accuracy ↑ | | | Error↓ | | Accuracy ↑ | | | Error↓ | | Accuracy ↑ | | | Error↓ | |
| | | | 5 | 10 | 45 | Mean | Med. | 5 | 10 | 25 | Mean | Med. | 1 | 5 | 10 | Mean | Med. |
| ICP Besl & McKay (1992) | - | | 31.7 | 55.6 | 99.6 | 10.4 | 8.8 | 7.5 | 19.4 | 74.6 | 22.4 | 20.0 | 8.4 | 24.7 | 40.5 | 32.9 | 14.1 |
| FPFH Rusu et al. (2009) | - | | 34.1 | 64.0 | 90.3 | 20.6 | 7.2 | 8.8 | 26.7 | 66.8 | 42.6 | 18.6 | 27.0 | 60.8 | 73.3 | 23.3 | 2.9 |
| SIFT Lowe (2004) | - | | 55.2 | 75.7 | 89.2 | 18.6 | 4.3 | 17.7 | 44.5 | 79.8 | 26.5 | 11.2 | 38.1 | 70.6 | 78.3 | 42.6 | 1.7 |
| SuperPoint DeTone et al. (2018) | - | | 65.5 | 86.9 | 96.6 | 8.9 | 3.6 | 21.2 | 51.7 | 88.0 | 16.1 | 9.7 | 45.7 | 81.1 | 88.2 | 19.2 | 1.2 |
| FCGF Choy et al. (2019) | - | ✓ | 70.2 | 87.7 | 96.2 | 9.5 | 3.3 | 27.5 | 58.3 | 82.9 | 23.6 | 8.3 | 52.0 | 78.0 | 83.7 | 24.4 | 0.9 |
| DGR Choy et al. (2020) | 3DMatch | ✓ | 81.1 | 89.3 | 94.8 | 9.4 | 1.8 | 54.5 | 76.2 | 88.7 | 18.4 | 4.5 | 70.5 | 85.5 | 89.0 | 13.7 | 0.4 |
| 3D MV Reg Gojcic et al. (2020) | 3DMatch | ✓ | 87.7 | 93.2 | 97.0 | 6.0 | 1.2 | 69.0 | 83.1 | 91.8 | 11.7 | 2.9 | 78.9 | 89.2 | 91.8 | 10.2 | 0.2 |
| REGTR Yew & Lee (2022) | 3DMatch | ✓ | 86.0 | 93.9 | 98.6 | 4.4 | 1.6 | 61.4 | 80.3 | 91.4 | 14.4 | 3.8 | 80.9 | 90.9 | 93.6 | 13.5 | 0.2 |
| UR&R El Banani et al. (2021) | 3DMatch | | 87.6 | 93.1 | 98.3 | 4.3 | 1.0 | 69.2 | 84.0 | 93.8 | 9.5 | 2.8 | 79.7 | 91.3 | 94.0 | 7.2 | 0.2 |
| UR&R(RGB-D) | 3DMatch | | 87.6 | 93.7 | 98.8 | 3.8 | 1.1 | 67.5 | 83.8 | 94.6 | 8.5 | 3.0 | 78.6 | 91.7 | 94.6 | 6.5 | 0.2 |
| UR&R(Supervised) | 3DMatch | ✓ | 92.3 | 95.3 | 98.2 | 3.8 | **0.8** | 77.6 | 89.4 | 95.5 | 7.8 | 2.3 | 86.1 | 94.0 | 95.6 | 6.7 | **0.1** |
| BYOC El Banani & Johnson (2021) | 3DMatch | | 66.5 | 85.2 | 97.8 | 7.4 | 3.3 | 30.7 | 57.6 | 88.9 | 16.0 | 8.2 | 54.1 | 82.8 | 89.5 | 9.5 | 0.9 |
| LLT Wang et al. (2022) | 3DMatch | | 93.4 | 96.5 | 98.8 | 2.5 | **0.8** | 76.9 | 90.2 | 96.7 | 5.5 | 2.2 | 86.4 | 95.1 | 95.8 | 4.6 | **0.1** |
| PointMBF Yuan et al. (2023) | 3DMatch | | 94.6 | 97.0 | 98.7 | 3.0 | **0.8** | 81.0 | 92.0 | 97.1 | 6.2 | **2.1** | 91.3 | 96.6 | 97.4 | 4.9 | **0.1** |
| F2M-Reg(Ours) | 3DMatch | | 96.3 | 98.7 | 99.7 | 1.8 | 0.9 | 81.9 | 94.7 | 98.5 | 4.2 | 2.4 | 91.4 | 97.9 | 98.6 | 3.1 | 0.1 |
| UR&R El Banani et al. (2021) | ScanNet | | 92.7 | 95.8 | 98.5 | 3.4 | 0.8 | 77.2 | 89.6 | 96.1 | 7.3 | 2.3 | 86.0 | 94.6 | 96.1 | 5.9 | **0.1** |
| UR&R(RGB-D) | ScanNet | | 94.1 | 97.0 | 99.1 | 2.6 | 0.8 | 78.4 | 91.1 | 97.3 | 5.9 | 2.3 | 87.3 | 95.6 | 97.2 | 5.0 | **0.1** |
| BYOC El Banani & Johnson (2021) | ScanNet | | 86.5 | 95.2 | 99.1 | 3.8 | 1.7 | 56.4 | 80.6 | 96.3 | 8.7 | 4.3 | 78.1 | 93.9 | 96.4 | 5.6 | 0.3 |
| LLT Wang et al. (2022) | ScanNet | | 95.5 | 97.6 | 99.1 | 2.5 | 0.8 | 80.4 | 92.2 | 97.6 | 5.5 | 2.2 | 88.9 | 96.4 | 97.6 | 4.6 | **0.1** |
| PointMBF Yuan et al. (2023) | ScanNet | | 96.0 | 97.6 | 98.9 | 2.5 | **0.7** | 83.9 | 93.8 | 97.7 | 5.6 | **1.9** | 92.8 | 97.3 | 97.9 | 4.7 | **0.1** |
| F2M-Reg(Ours) | ScanNet | | 97.6 | 99.1 | 99.8 | 1.4 | 0.8 | 85.5 | 95.8 | 98.8 | 3.7 | 2.1 | 93.1 | 98.4 | 98.9 | 2.9 | **0.1** |

Table 2: **Pairwise registration on ScanNet with a 50 frames apart setting.**

| | Train Set | Rotation(°) | | | | | Translation(cm) | | | | | Chamfer(mm) | | | | |
|---|---|---|---|---|---|---|---|---|---|---|---|---|---|---|---|---|
| | | Accuracy ↑ | | | Error↓ | | Accuracy ↑ | | | Error↓ | | Accuracy ↑ | | | Error↓ | |
| | | 5 | 10 | 45 | Mean | Med. | 5 | 10 | 25 | Mean | Med. | 1 | 5 | 10 | Mean | Med. |
| PointMBF | 3DMatch | 59.3 | 62.5 | 76.6 | 25.5 | 3.2 | 34.2 | 47.9 | 61.6 | 51.2 | 8.2 | 42.9 | 55.8 | 60.2 | 103.1 | 1.3 |
| F2M-Reg(Ours) | 3DMatch | 72.6 | 81.1 | 91.1 | 12.5 | 2.2 | 44.6 | 65.9 | 78.5 | 28.5 | 5.8 | 56.1 | 73.6 | 77.1 | 72.0 | 0.7 |
| PointMBF | ScanNet | 60.4 | 68.2 | 79.9 | 19.2 | 2.3 | 40.0 | 54.3 | 66.9 | 38.1 | 6.0 | 48.9 | 61.5 | 65.8 | 85.8 | 0.7 |
| F2M-Reg(Ours) | ScanNet | 77.4 | 84.5 | 92.5 | 15.5 | 1.9 | 50.0 | 70.6 | 82.1 | 30.1 | 5.0 | 61.5 | 77.6 | 80.9 | 73.8 | 0.5 |

## 4.2 QUANTITATIVE RESULTS

We claim our experimental comparison with existing methods in Table. 1 and Table. 2. Our framework is warmed up on the Sim-RGBD dataset and fine-tuned on ScanNet(Dai et al., 2017) and 3DMatch(Zeng et al., 2017), respectively.

**Evaluate on 20 frames apart.** As shown in Table 1, using the same training and test sets, our proposed method achieves significant improvements across nearly all metrics, surpassing even the previous state-of-the-art method, PointMBF (Yuan et al., 2023). While the frame-to-frame optimization framework leverages the RGB-D sequence to train the network in an unsupervised manner, it falls short of fully utilizing the scene's global contextual information. In contrast, our frame-to-model approach proves to be a more effective training strategy for the registration model. The experimental results validate our hypothesis that frame-to-model manner plays a crucial role in RGB-D point cloud registration. Additionally, the warming-up process provides the registration model with better initialization, allowing it to generate more accurate initial poses for the fine-tuning stage. This will be further demonstrated in our ablation studies.

**Evaluate on 50 frames apart.** Since the test with a 20 frame apart is not enough to show the capability of our method with a large overlap, we separately compared our method with the current state-of-the-art method with a 50 frame apart setting in Table. 2. It is evident that our method achieves higher accuracy even with a smaller overlap.

## 4.3 ABLATION STUDY

To assess the contribution of various components in our pipeline, we conduct several ablation studies on the real-world datasets ScanNet (Dai et al., 2017) and 3DMatch (Zeng et al., 2017). In the following experiments, unless otherwise noted, the model is initially warmed up on Sim-RGBD.

**Comparison on rendering strategies.** The previous method generates the supervised signal by taking two frames as input, rasterizing the RGB and depth pictures of the other frame separately and making a loss with the GT image. In our pipeline, we replace the traditional point cloud rasterization

Table 3: **Ablation on training strategy.** *F2F* refers to the pairwise training strategy of rendering RGB and depth images from the aligned point cloud. *F2M* denotes the training the registration model based on neural rendering approach, where we optimize the input global pose through neural rendering. The optimized pose is then used to guide the supervision of the registration model.

| | | Accuracy ↑ | | | Error↓ | | Accuracy ↑ | | | Error↓ | | Accuracy ↑ | | | Error↓ | |
|---|---|---|---|---|---|---|---|---|---|---|---|---|---|---|---|---|
| | Train Set | 5 | 10 | 45 | Mean | Med. | 5 | 10 | 25 | Mean | Med. | 1 | 5 | 10 | Mean | Med. |
| F2F | ScanNet | 75.2 | 82.5 | 90.3 | **14.0** | 2.0 | 47.4 | 68.3 | 80.5 | **30.0** | 5.4 | 58.8 | 76.5 | 79.4 | **69.2** | 0.6 |
| F2M | ScanNet | **77.4** | **84.5** | **92.5** | 15.5 | **1.9** | **50.0** | **70.6** | **82.1** | 30.1 | **5.0** | **61.5** | **77.6** | **80.9** | 73.8 | **0.5** |

Table 4: **Ablation on warming up module.** *Warm up* denotes the process of training the registration model on the Sim-RGBD dataset.

| | | Rotation(°) | | | | | Translation(cm) | | | | | Chamfer(mm) | | | | |
|---|---|---|---|---|---|---|---|---|---|---|---|---|---|---|---|---|
| | | Accuracy ↑ | | | Error↓ | | Accuracy ↑ | | | Error↓ | | Accuracy ↑ | | | Error↓ | |
| Warm up | Train Set | 5 | 10 | 45 | Mean | Med. | 5 | 10 | 25 | Mean | Med. | 1 | 5 | 10 | Mean | Med. |
| | 3DMatch | 59.0 | 71.3 | 86.3 | 18.6 | 3.5 | 30.4 | 52.3 | 68.0 | 42.2 | 9.2 | 43.2 | 62.1 | 66.5 | 97.0 | 1.6 |
| ✓ | 3DMatch | **72.6** | **81.1** | **91.1** | **12.5** | **2.2** | **44.6** | **65.9** | **78.5** | **28.5** | **5.8** | **56.1** | **73.6** | **77.1** | **72.0** | **0.7** |
| | ScanNet | 74.4 | 82.8 | 92.3 | **10.8** | 2.1 | 46.8 | 67.9 | 80.4 | **25.4** | 5.5 | 58.5 | 75.5 | 79.0 | **67.1** | 0.6 |
| ✓ | ScanNet | **77.4** | **84.5** | **92.5** | 15.5 | **1.9** | **50.0** | **70.6** | **82.1** | 30.1 | **5.0** | **61.5** | **77.6** | **80.9** | 73.8 | **0.5** |

step with neural rendering approach based on the neural implicit field. To demonstrate the effectiveness of our rendering strategy, we conduct an ablation study on this two rendering approaches.

As shown in Table 3, under the same conditions and initialization, the neural rendering approach outperforms their point cloud rasterization method. This result supports our claim that constructing a neural filed using multiple RGB-D images more effectively exploits both photometric and geometric consistency within the scene. Compared to the point cloud rasterization approach, our method provides a stronger supervisory signal to the registration model, leading to better performance.

**Effectiveness of warming up module.** In the preceding discussion, constructing an accurate neural implicit field relies on having a reasonably accurate initial camera pose. We believe that performing warming up on a synthetic dataset can assist the registration model in providing a better camera pose when operating in real-world scenes. We consider the warming up process to be essential, as the absence of this stage significantly degrades the final performance. Consequently, we conduct an ablation study on the presence or absence of warming up.

The results are shown in Table. 4. Our findings indicate a substantial decrease in performance when the warming up operation is omitted. This aligns with our hypothesis that without a sufficiently good initial camera pose, the effectiveness of frame-to-model optimization manner in camera pose optimization diminishes, and in some cases, may even result in no improvement at all.

**Effectiveness of fine-tuning module.** To demonstrate the necessity of fine-tuning module, which use frame-to-model manner for pose optimization, we conduct this ablation study. We test four models on the ScanNet dataset, each representing a different training strategy: a registration model obtained after warming up, a registration model fine-tuned using the pose generated by RANSAC, a registration model fine-tuned using the pose optimized with 20 tracking iterations, and a registration model fine-tuned using the pose optimized with 100 tracking iterations.

The results are presented in Table 5. The four rows correspond to the models described above and the results reveal two key findings: First, fine-tuning on real-world scenes is essential, as the last three rows show significant improvements compared to the first row. Second, the quality of the poses used to generate supervisory signals is crucial to the model's performance, as demonstrated by the progressively stronger results of the registration model with the iterations of tracking increasing.

## 4.4 QUALITATIVE RESULTS

Fig. 4 provides a gallery of the correspondence results of the model trained on ScanNet with PointMBF and F2M-Reg. The frame-to-model optimization enables the registration model to perform better on scenes with smaller overlaps (1[st] row) and more dramatic changes in lighting conditions (2[nd] and 3[rd] rows) between relative frames.

Table 5: **Ablation on fine-tuning module.** *FT* refers to fine-tuning registration model on the train set, as described in Sec.3.3. $N_t$ denotes the number of iterators for tracking. Blanks in the *FT* column indicate that no fine-tuning operation is performed while blanks in the $N_t$ column signify that no neural implicit field is constructed in those cases.

| FT | $N_t$ | Train Set | Rotation(°) | | | | | Translation(cm) | | | | | Chamfer(mm) | | | | |
| | | | Accuracy ↑ | | | Error↓ | | Accuracy ↑ | | | Error↓ | | Accuracy ↑ | | | Error↓ | |
| | | | 5 | 10 | 45 | Mean | Med. | 5 | 10 | 25 | Mean | Med. | 1 | 5 | 10 | Mean | Med. |
| | | | 71.3 | 78.6 | 87.4 | 15.8 | 2.0 | 46.9 | 65.5 | 76.3 | 34.5 | 5.4 | 57.5 | 72.2 | 75.0 | 77.7 | 0.6 |
| ✓ | | ScanNet | 76.0 | 84.3 | 92.7 | 10.3 | 2.0 | 46.8 | 69.4 | 81.8 | 23.8 | 5.4 | 59.4 | 77.1 | 80.6 | 61.5 | 0.6 |
| ✓ | 20 | ScanNet | 76.7 | 84.4 | **92.9** | **10.0** | 1.9 | 48.4 | 69.7 | 81.8 | **22.7** | 5.2 | 60.7 | 77.1 | 80.2 | **60.4** | 0.6 |
| ✓ | 100 | ScanNet | **77.4** | **84.5** | 92.5 | 15.5 | **1.9** | **50.0** | 70.6 | **82.1** | 30.1 | **5.0** | **61.5** | **77.6** | **80.9** | 73.8 | **0.5** |

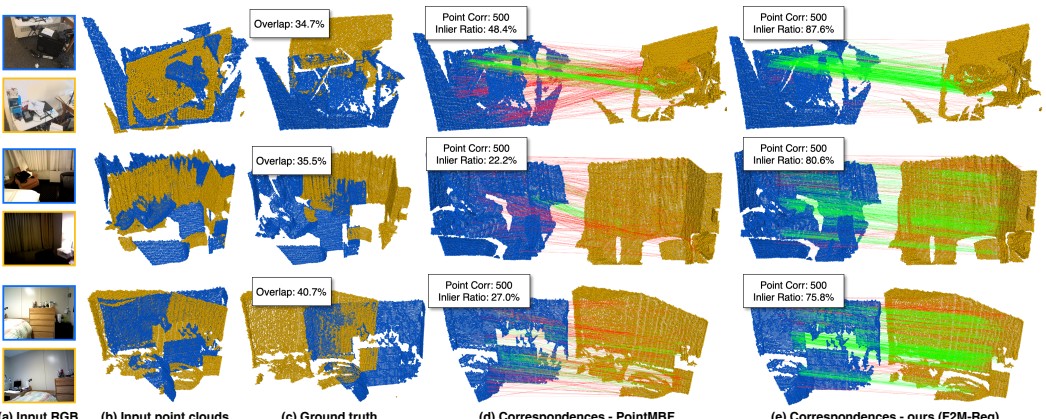

Figure 4: **Correspondences results with PointMBF and F2M-Reg.** The first row shows that F2M-Reg outperforms when the input point clouds have a low overlap ratio. The subsequent rows illustrate that even under significant lighting changes, which adversely affect other methods, our approach continues to perform effectively.

## 5 CONCLUSION

We present F2M-Reg, a frame-to-model optimization framework for unsupervised RGB-D registration. By constructing a neural implicit field and enforcing photometric and geometric consistency between input and rendered frames, we optimize the estimated poses from registration model. This design significantly enhances robustness against lighting changes, geometric occlusions, and reflective materials. Additionally, We implement a warming up mechanism on a synthetic dataset to initialize neural field optimization effectively. Extensive experiments on two benchmarks demonstrate effectiveness of our methods. We believe the unsupervised learning framework with frame-to-model optimization holds promise for future 3D vision tasks like localization and reconstruction.

## 6 LIMITATION

Despite achieving state-of-the-art performance, our method has some limitations. First, it cannot be directly applied to outdoor scenes. Excessive depth variations between foreground and background can lead to incorrect subscene construction. Also, encoding a large open scene within a single neural implicit field poses significant challenges. Second, the method requires a relatively long training time due to the optimization of the neural field.

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

## A  APPENDIX

In the appendix, we organize the details as follows. First, in Sec. B, we present the specifics of our implementations. Second, Sec. C introduces additional experiments that focus on the effectiveness of loss functions, comparision with supervised learning based on SLAM, and the convergence of our framework. Finally, more qualitative results are presented in Sec. D.

# B IMPLEMENTATION DETAILS OF F2M-REG

## B.1 NEURAL IMPLICIT FIELD OPTIMIZATION

For each sequence, we initially conduct 200 iterations of mapping using the first frame to establish the initialization of the neural implicit field. Subsequently, during the training phase, we input both the current frame and the neural implicit field $\mathcal{M}$. We maintain $\mathcal{M}$ fixed while optimizing the camera parameters of the untracked pose. Specifically, we randomly select $N_t = 1024$ pixels from the current frame. For each ray, we uniformly sample $M_c = 32$ points between the near and far bounds. Additionally, we sample an extra $M_f = 21$ depth-guided points evenly within the range $[d - d_s, d + d_s]$, where $d$ represents the depth and $d_s = 0.25$ denotes a small offset. During experimentation, it was noted that employing the same number of optimization rounds as in Co-SLAM (Wang et al., 2023) often yielded suboptimal untracked poses across most scenes. This challenge arises due to the larger inter-frame distances present in our data compared to scenes encountered in previous Neural SLAM tasks. Hence, we conduct 100 iterations for tracking to mitigate this issue.

Subsequently, the tracked pose is utilized in the mapping stage. Upon the initial addition of each frame to the mapping stage, 5% of its pixels are incorporated into the maintained pixel bank. During the mapping phase, 2048 pixels are randomly selected from the pixel bank, and rays are generated to participate in the training process. The subsequent procedure mirrors that of the tracking section, except that the optimized parameters $\Psi = \{\theta, \hat{T}_{i-2}, \hat{T}_{i-1}, \tilde{T}_i\}$ consist of the neural implicit field $\mathcal{M}$ and the camera poses in the batch.

Our sub-scene representation comprises a $L = 16$ level hash grid $\mathcal{V}_\alpha = \mathcal{V}_{\alpha\,l=1}^{l\ \ L}$, with 16 bins oneblob for each dimension. The color and Signed Distance Function (SDF) are encoded by two 2-layer MLPs with 32 hidden units and a 15-dimensional geometric feature. The boundaries of our sub-scene are confined within the following ranges along the xyz-axes: (-3, 7), (-5, 5), (-4, 4). Regarding the learning rates, we utilize $\eta_t = 0.002$ for tracking, and $\eta_f = 0.01$, $\eta_d = 0.01$, and $\eta_p = 0.0005$ for the feature grid, decoder, and pose optimizer, respectively.

In both of the above stages, we minimize four different losses introduced in Co-SLAM(Wang et al., 2023). They include (1) two rendering losses $\mathcal{L}_{RGB}$ and $\mathcal{L}_{depth}$ for minimizing errors between ground truth RGB/depth image $\hat{C}_p/\hat{D}_p$ and rendered RGB/depth image $C_p/D_p$:

$$\mathcal{L}_{RGB} = \frac{1}{|P|} \sum_{p \in P} (C_p - \hat{C}_p)^2, \mathcal{L}_{depth} = \frac{1}{|P|} \sum_{p \in P} (D_p - \hat{D}_p)^2, \tag{3}$$

where $P$ represents sampled image pixels. (2) an SDF loss $\mathcal{L}_{sdf}$ to enhance the consistency of the SDF field:

$$\mathcal{L}_{sdf} = \frac{1}{|P|} \sum_{p \in P} \frac{1}{|S_p^{tr}|} \sum_{s \in S_p^{tr}} (D_s - \hat{D}_s)^2, \tag{4}$$

$S_p^{tr}$ represents whose signed distance function (SDF) is not truncated along the viewing ray of pixel $p$, and $D_s/\hat{D}_s$ denote their predicted/ground-truth SDF values. (4) For those sampled points distant from the observed surface, a free-space loss $\mathcal{L}_{fs}$ is applied to enforce their predicted SDF to be truncation distance $d_{tr}$:

$$\mathcal{L}_{fs} = \frac{1}{|P|} \sum_{p \in P} \frac{1}{|S_p^{fs}|} \sum_{s \in S_p^{fs}} (D_s - d_{tr})^2. \tag{5}$$

(5) An additional regularization on the interpolated features $\mathcal{V}_\alpha(x)$ in order to decrease the noisy in reconstruction.

$$\mathcal{L}_{smooth} = \frac{1}{\mathcal{V}} \sum_{x \in |\mathcal{V}|} (\Delta_x^2 + \Delta_y^2 + \Delta_z^2) \tag{6}$$

where $\mathcal{V}$ denotes the grid and $\Delta_{xyz} = \mathcal{V}_\alpha(x + \epsilon_{xyz}) - \mathcal{V}_\alpha(x)$ The weights of each loss are $\lambda_{RGD} = 5.0$, $\lambda_{depth} = 0.1$, $\lambda_{sdf} = 1000$, $\lambda_{fs} = 10$, $\lambda_{smooth} = 0.001$.

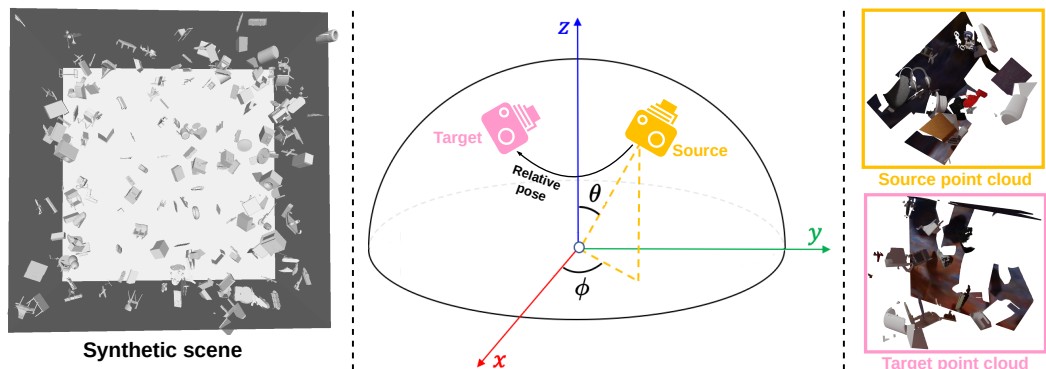

Figure 5: **Demonstration of Sim-RGBD dataset.** The entire scene is depicted in the left figure. Camera sampling is illustrated in the middle figure. Initially, we sample the position of the first camera based on a specified pitch angle $\theta$ and yaw angle $\phi$, with (0, 0, 0) as the viewpoint, forming the camera's view direction. The position of the second camera is derived from the transformation of the first camera position, which is obtained from a Gaussian distribution. The right figure showcases the point cloud with color extracted from the scene.

## B.2 REGISTRATION MODEL TRAINING

Supervisory signals for the feature extractor are generated when the current batch is fixed. These signals are generated by optimized pose $(\hat{T}_i, \hat{T}_j)$, corresponding point clouds $(X_i, X_j)$, and corresponding features $(F_i, F_j)$ within the current batch. Specifically, the process begins by deriving a relative pose using the global pose optimized within neural implicit representation. For one relative pose $\Delta T_{i-1,i}$, correspondences between two point clouds are identified using a specified threshold $\tau$, which can be formulated as $\mathcal{C}^* = \{(p_{i-1}, p_i) \mid ||T_{i-1,i}p_{i-1} - p_i|| < \tau\}$. The correspondences and corresponding feature pairs $\{(F^{p_{i-1}}, F^{p_i}) \mid (p_{i-1}, p_i) \in \mathcal{C}^*\}$ are utilized to compute the loss.

## B.3 GEOMETRIC FITTING

Given 512 input correspondences, $\mathcal{C} = \{(\mathbf{p}_i, \mathbf{q}_i) \mid \mathbf{p}_i \in \mathbf{X}, \mathbf{q}_i \in \mathbf{Y}\}$, we randomly sample $t = 10$ subsets, each containing $l = 20\%$ of the total correspondences. For each subset, a candidate transformation $T$ is estimated by solving a Weighted Procrustes problem (Besl & McKay, 1992). The candidate transformation $T^*$ that minimizes the error $E(C, T^*)$ is retained. Additionally, during the testing phase, we increase $t$ to 100 and reduce $l$ to $5\%$ to achieve better RANSAC results while limiting computational costs.

## B.4 SIM-RGBD

To clarify how we sample the appropriate camera poses, we visualize the process of sampling two camera poses in Fig. 5.

Table 6: Implementation details of our F2M-Reg

(a) Same setting as PointMBF

| Momentum | 0.9 |
|---|---|
| Optimizer | Adam |
| Image size | 128*128 |
| Feature dimension | 32 |
| $K_{v2g}, K_{g2v}$ for training | $K_{v2g} = 16, K_{g2v} = 1$ |
| $K_{v2g}, K_{g2v}$ for test | $K_{v2g} = 32, K_{g2v} = 1$ |

(b) Changes in F2M-Reg

| Batch size | 4 |
|---|---|
| Normalization in ResNet | GroupNorm |
| Normalization in fusion | GroupNorm |
| Group of channels | 32 |
| low's ratio | False |
| Number of correspondence $k$ | 256 |

## B.5 SETTINGS

We adopt PointMBF (Yuan et al., 2023) as our registration model and utilize several of its settings, such as data processing and learning rate. On the software side, our code is built using PyTorch and PyTorch3D (Ravi et al., 2020). On the hardware side, we train our network using an Nvidia GeForce RTX 3090Ti GPU with 24GB of memory, paired with an Intel® Core™ i9-12900K @ 3.9GHz × 16 and 32GB of RAM. To ensure a fair comparison, we adhere to the same training schemes as PointMBF, including data processing and other configurations. Table 6 provides further details on the similarities and differences between our approach and PointMBF. Specifically, Table 6a outlines the shared settings, while Table 6b highlights the differing configurations.

## B.6 TIME EFFICIENCY

Table 7: Runtime analysis

|  | Time(ms) |
|---|---|
| Feature Extraction | $79.87 \pm 29.10$ |
| Correspondence Estimation | $35.57 \pm 10.36$ |
| Geometric Fitting | $10.32 \pm 9.20$ |
| Loss Computing | $28.97 \pm 10.38$ |
| Backward | $202.34 \pm 164.08$ |
| Tracking(Just for training) | $20.10 \pm 4.41$ |
| Mapping(Just for training) | $25.06 \pm 2.59$ |

The time for our pipeline was reported in Table 7. Our method increases the training time, focusing on the tracking and mapping stage, i.e., we have to spend time on optimizing neural implicit field compared to rasterization of the point cloud. But considering the excellent performance of the frame-to-model set of optimization frameworks, we think these time overheads are meaningful.

# C ADDITIONAL EXPERIMENTS

## C.1 EFFECTIVENESS ON LOSS

Our work incorporates the circle loss (Sun et al., 2020; Huang et al., 2021) and correspondence loss into the training process. We conduct a comprehensive Ablation study to elucidate the significance of these two losses within the entire pipeline.

The correspondence loss $L_{corr}$ (1) is formalized in equation 7. In the context, $\mathcal{C} = \{(\mathbf{p}_i, \mathbf{q}_i) \mid \mathbf{p}_i \in \mathbf{X}, \mathbf{q}_i \in \mathbf{Y}\}$ denotes the correspondences selected based on the cosine similarity of the features of corresponding two points. We choose the top 256 pairs of correspondences and use the relative optimized pose $\Delta \hat{T} = [\hat{R} \mid \hat{t}]$ to calculate the loss. The weights $w_i$ range from 0 to 1, and are derived from the cosine similarity values of the two point features.

$$L_{corr} = \sum_{(\mathbf{p}_i, \mathbf{q}_i) \in \mathcal{C}} w_i \|\hat{R}\mathbf{p}_i + \hat{t} - \mathbf{q}_i\| \tag{7}$$

To better supervise the point-wise descriptors, we also follow (Huang et al., 2021) and employ the circle loss. Considering the correspondence $\mathcal{C} = \{(\mathbf{p}_i, \mathbf{q}_i) \mid \mathbf{p}_i \in \mathbf{X}, \mathbf{q}_i \in \mathbf{Y}\}$ and the optimized pose $\hat{T}$. We compute, for each point in $\mathbf{X}$ the distance to all points in $\mathbf{Y}$. Pairs of points with a

Table 8: **Ablation on loss.** *Corr* denotes the correspondence loss. *Circle* denotes the circle loss. All the blank (1st row) means the registration model was only bootstrapped on the Sim-RGBD dataset without fintuning.

| Corr | Circle | Train Set | Rotation(°) | | | | | Translation(cm) | | | | | Chamfer(mm) | | | | |
|---|---|---|---|---|---|---|---|---|---|---|---|---|---|---|---|---|---|
| | | | Accuracy ↑ | | | Error↓ | | Accuracy ↑ | | | Error↓ | | Accuracy ↑ | | | Error↓ | |
| | | | 5 | 10 | 45 | Mean | Med. | 5 | 10 | 25 | Mean | Med. | 1 | 5 | 10 | Mean | Med. |
| | | | 71.3 | 78.6 | 87.4 | 15.8 | 2.0 | 46.9 | 65.5 | 76.3 | 34.5 | 5.4 | 57.5 | 72.2 | 75.0 | 77.7 | 0.6 |
| ✓ | | ScanNet | 71.0 | 77.7 | 86.7 | 18.8 | 2.0 | 48.7 | 66.0 | 75.8 | 34.7 | 5.2 | 58.1 | 72.2 | 75.0 | 83.8 | 0.6 |
| | ✓ | ScanNet | 77.1 | 84.3 | **92.7** | **10.3** | **1.9** | 49.1 | 70.2 | 81.6 | **24.0** | 5.1 | 61.1 | 77.3 | 80.4 | **62.7** | **0.5** |
| ✓ | ✓ | ScanNet | **77.4** | **84.5** | 92.5 | 15.5 | **1.9** | **50.0** | 70.6 | **82.1** | 30.1 | **5.0** | **61.5** | **77.6** | **80.9** | 73.8 | **0.5** |

distance less than $r_p$ are treated as positive samples $\epsilon_{pos}$, while those greater than $r_s$ are treated as negative samples $\epsilon_{neg}$. The circle loss from $\mathbf{X}$ is formalized in equation 8.

$$L_{circle}^{\mathbf{X}} = \frac{1}{n} \sum_{i=1}^{n} log[1 + \sum_{j \in \epsilon_{pos}} e^{\beta_{pos}^j (d_i^j - \Delta_{pos})} \cdot \sum_{k \in \epsilon_{neg}} e^{\beta_{neg}^k (\Delta_{neg} - d_i^k)}] \tag{8}$$

where $n$ is the number of the points in $\mathbf{X}$, $d_i^j = \|f_{p_i} - f_{q_i}\|$ denotes the L2 distance of the corresponding point features and $\Delta_{pos}$, $\Delta_{neg}$ are positive and negative margins. The weights $\beta_{pos}^j = \gamma(d_i^j - \Delta_{pos})$ and $\beta_{neg}^k = \gamma(\Delta_{neg} - d_i^k)$ are computed for each correspondence. The margin hyper-parameters are set to $\Delta_{pos} = 0.1$ and $\Delta_{neg} = 1.4$. For the circle loss $L_{circle}^{\mathbf{Y}}$ goes the same. The final circle loss $L_{circle} = (L_{circle}^{\mathbf{X}} + L_{circle}^{\mathbf{Y}})/2$.

The outcomes are presented in Table 8. It is evident from the results that both losses are instrumental in enhancing the performance of our registration framework during the finetuning phase. Regarding the nature of the losses, the circle loss facilitates the accurate recognition of correspondences by the registration model, whereas the correspondence loss aids in adjusting the weighting of identified correspondences. Furthermore, the experiment confirms that the concurrent utilization of these two losses contributes to further advancements in our registration model.

## C.2 COMPARISON WITH SUPERVISED LEARNING BASED ON SLAM.

For sequential data, it is intuitive to obtain the pose of each frame quickly through reconstruction pipelines like SLAM. The pose reconstructed by SLAM can be used to supervise the training of a registration model. However, our approach can further enhance the performance of a registration model trained on SLAM-reconstructed poses. To demonstrate the effectiveness of our optimization framework, we designed this ablation experiment. We selected ROSEFusion (Zhang et al., 2021) as the SLAM algorithm for the experiment.

We randomly divide the 1,045 ScanNet training scenes into two groups: 300 scenes and 745 scenes. Using ROSEFusion, we perform the reconstruction on the 300 scenes, and the resulting poses were used to train a registration model from scratch. We then fine-tune the registration model using our frame-to-model optimization framework on the next 745 scenes. Finally, we test all the registation model on ScanNet testing scenes.

The results, as shown in Table 9, indicate that the performance of the registration model improves as the number of training scenes increases, demonstrating the optimization strength of our method. Notably, our bootstrapping module is flexible and can accommodate different datasets. When the quality of the synthetic dataset or the accuracy of the reconstruction from real-world scenes improves, these factors contribute to a better initialization of the registration model. Given a highly expressive model, our framework can leverage these improvements to provide a stronger initialization, further enhancing the registration model's performance.

## C.3 CONVERGENCE

We conduct quantitative experiments to assess the convergence of our framework on the Scan-Net (Dai et al., 2017), with the registration model fine-tuned on the 3DMatch (Zeng et al., 2017).

Table 9: **Comparison with supervised learning based on SLAM.** $N_s$ indicates the number of scenes used for training. The first row shows the results of using the reconstructed poses from ROSEFusion on 300 scenes to supervise the registration model, which was then tested on ScanNet. The subsequent rows display the outcomes of increasing the number of training scenes from the first row and further optimizing the results using our pipeline.

| $N_s$ | Rotation(°) | | | | | Translation(cm) | | | | | Chamfer(mm) | | | | |
|---|---|---|---|---|---|---|---|---|---|---|---|---|---|---|---|
| | Accuracy ↑ | | | Error↓ | | Accuracy ↑ | | | Error↓ | | Accuracy ↑ | | | Error↓ | |
| | 5 | 10 | 45 | Mean | Med. | 5 | 10 | 25 | Mean | Med. | 1 | 5 | 10 | Mean | Med. |
| 300 (ROSEFusion) | 70.5 | 79.6 | 90.4 | 12.5 | 2.3 | 42.9 | 64.5 | 77.4 | 28.7 | 5.9 | 55.0 | 72.2 | 75.7 | 73.1 | 0.7 |
| +200 (ROSEFusion+Ours) | 71.1 | 80.6 | 91.2 | 12.0 | 2.3 | 42.5 | 64.4 | 77.8 | 27.5 | 6.0 | 54.7 | 72.5 | 76.1 | 70.8 | 0.7 |
| +400 (ROSEFusion+Ours) | 71.6 | 80.6 | 90.9 | 11.4 | 2.2 | 43.3 | 64.7 | 77.7 | 27.0 | 5.9 | 55.2 | 72.7 | 76.2 | 69.2 | 0.7 |
| +600 (ROSEFusion+Ours) | **73.2** | 81.9 | 91.6 | 11.1 | **2.1** | **45.8** | 66.5 | **79.7** | 25.6 | **5.5** | **57.4** | 74.4 | **78.1** | 65.9 | 0.6 |
| +745 (ROSEFusion+Ours) | **73.2** | **82.1** | **91.7** | **10.7** | **2.1** | 45.3 | **66.6** | 79.6 | **24.7** | **5.5** | 57.1 | **74.5** | 78.0 | **63.0** | **0.6** |

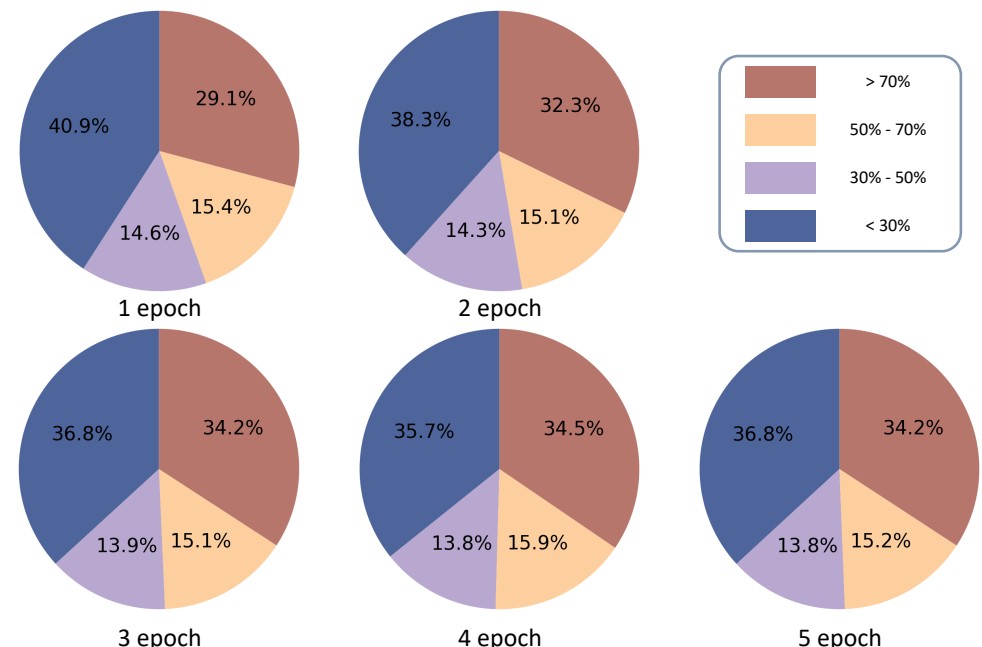

Figure 6: **Variation of Inlier Ratio Across Training Epochs.** The pie chart illustrates the distribution of inlier ratios across different ranges, with each range represented by distinct colored blocks. Larger blocks correspond to a higher number of frame pairs that fall within the respective inlier ratio range on the test dataset. This data was obtained by fine-tuning the registration model on the 3DMatch dataset Zeng et al. (2017) and testing it on the ScanNet test set Dai et al. (2017), using frame pairs spaced 50 frames apart.

Figure 6 and table 10 depict the evolution on the ScanNet test set as the number of epochs progresses. We observe an initial increase in the inlier ratio until epoch 4, followed by oscillations. Again this phenomenon appears in the performance of the pose. These findings underscore the convergence of our framework. Our framework comprises two integral components: the registration model and the neural implicit field. They synergistically reinforce each other, wherein the enhanced performance of the registration model contributes to improved quality of the global pose input for neural field. Consequently, this enhances the reconstruction quality of neural field. The improved neural field quality facilitates more accurate and efficient optimization of the untracked poses, thereby providing a more effective gradient to refine the preceding registration model.

Table 10: **Table of performance effects of training different epoch registation.** *Epoch* indicates the number of epochs that the registration model has been trained on 3DMatchZeng et al. (2017).

| Epoch | Train Set | Rotation(°) | | | | | Translation(cm) | | | | | Chamfer(mm) | | | | |
| | | Accuracy ↑ | | | Error↓ | | Accuracy ↑ | | | Error↓ | | Accuracy ↑ | | | Error↓ | |
| | | 5 | 10 | 45 | Mean | Med. | 5 | 10 | 25 | Mean | Med. | 1 | 5 | 10 | Mean | Med. |
| 1 | 3DMatch | 71.2 | 79.9 | 90.3 | 13.6 | 2.3 | 42.4 | 63.9 | 77.3 | 30.8 | 6.2 | 54.5 | 72.3 | 75.8 | 75.1 | 0.8 |
| 2 | 3DMatch | 72.6 | 81.1 | 91.1 | 12.5 | **2.2** | 44.6 | 65.9 | 78.5 | 28.5 | **5.8** | 56.1 | 73.6 | 77.1 | 72.0 | **0.7** |
| 3 | 3DMatch | 73.0 | 81.9 | 91.4 | 12.4 | 2.3 | 44.0 | 65.7 | 79.2 | 29.0 | 6.2 | 56.2 | 73.9 | 77.8 | 71.8 | 0.8 |
| 4 | 3DMatch | **73.5** | **82.2** | **91.6** | **11.9** | 2.2 | **44.7** | **66.4** | **79.7** | **26.8** | **5.8** | **56.9** | **74.5** | **78.2** | **69.0** | **0.7** |
| 5 | 3DMatch | 73.0 | 81.7 | 91.4 | 12.2 | 2.4 | 44.5 | 65.9 | 79.4 | 28.2 | 6.4 | 56.6 | 74.0 | 77.7 | 71.5 | 0.8 |

## C.4 ABLATION ON DIFFERENT MODULE COMBINATIONS

We have separately studied the improvements from each component in Tab.2, Tab.3, Tab.4 and Tab.5 in the main paper. Here, we reorganize the results in Tab 11 below for a better understanding. Warm up refers to the synthetic warm-up, F2M refers to frame-to-model optimization, and F2F refers to the frame-to-frame optimization. Applying the synthetic warm-up mechanism (line 2) and frame-to-model optimization (line 3) independently both result in significant improvements over the frame-to-

Table 11: **Ablation on different module combination.**

| Warm up | F2F | F2M | Rotation(°) | | | | | Translation(cm) | | | | | Chamfer(mm) | | | | |
| | | | Accuracy ↑ | | | Error↓ | | Accuracy ↑ | | | Error↓ | | Accuracy ↑ | | | Error↓ | |
| | | | 5 | 10 | 45 | Mean | Med. | 5 | 10 | 25 | Mean | Med. | 1 | 5 | 10 | Mean | Med. |
| | ✓ | | 60.4 | 68.2 | 79.9 | 19.2 | 2.3 | 40.0 | 54.3 | 66.9 | 38.1 | 6.0 | 48.9 | 61.5 | 65.8 | 85.8 | 0.7 |
| ✓ | | | 71.3 | 78.6 | 87.4 | 15.8 | 2.0 | 46.9 | 65.5 | 76.3 | 34.5 | 5.4 | 57.5 | 72.2 | 75.0 | 77.7 | 0.6 |
| | | ✓ | 74.4 | 82.8 | 92.3 | **10.8** | 2.1 | 46.8 | 67.9 | 80.4 | **25.4** | 5.5 | 58.5 | 75.5 | 79.0 | **67.1** | 0.6 |
| ✓ | ✓ | | 75.2 | 82.5 | 90.3 | 14.0 | 2.0 | 47.4 | 68.3 | 80.5 | 30.0 | 5.4 | 58.8 | 76.5 | 79.4 | 69.2 | 0.6 |
| ✓ | | ✓ | **77.4** | **84.5** | **92.5** | 15.5 | **1.9** | **50.0** | **70.6** | **82.1** | 30.1 | **5.0** | **61.5** | **77.6** | **80.9** | 73.8 | **0.5** |

Table 12: **Pairwise registration on TUM RGB-D with a 50 frames apart setting.**

| | Rotation(°) | | | | | Translation(cm) | | | | | Chamfer(mm) | | | | |
| | Accuracy ↑ | | | Error↓ | | Accuracy ↑ | | | Error↓ | | Accuracy ↑ | | | Error↓ | |
| | 5 | 10 | 45 | Mean | Med. | 5 | 10 | 25 | Mean | Med. | 1 | 5 | 10 | Mean | Med. |
| PointMBF | 85.9 | **97.9** | **100.0** | 2.5 | 1.5 | 66.5 | 84.3 | 98.4 | 5.1 | 3.1 | 69.6 | 86.4 | 91.6 | 2.5 | 0.4 |
| F2M-Reg | **95.3** | 96.9 | **100.0** | **1.8** | **1.1** | **78.0** | **94.8** | **99.5** | **3.7** | **2.6** | **79.1** | **95.3** | **96.9** | **1.1** | **0.3** |

frame baseline (line 1), highlighting the strong effectiveness of these designs. Moreover, combining both mechanisms further enhances performance, achieving an improvement of 6 percentage points over the baseline-only model and 3 percentage points over the frame-to-model-only model across most metrics.

## C.5 COMPARISON ON TUM RGB-D

We conduct a comparison with PointMBFYuan et al. (2023) and F2M-Reg on TUM RGB-D, which has more accurate and high-resolution RGB-D streams. The two registration models in Tab. 12 are trained on ScanNet and tested on TUM RGB-D with a 50 frames apart setting. We find that our method surpasses PointMBF by 9.4 percent point on Rotation Accuracy@5°, 11.5 percent point on Translation Accuracy@5cm, and 9.5 percent point on Chamfer Accuracy@1cm. These results have proven the strong generality of method to new datasets.

## D QUALITATIVE VISUALIZATION

In this section, we present more detailed visualization results in Fig 7 for both our method and PointMBF(Yuan et al., 2023). We visualize the inputs and the final alignment outcomes. In dataset selection, we deliberately choose scenes with minimal overlap and significant lighting variations. From the visualization results, our method exhibits several advantages. This observation further supports the superiority of the frame-to-model approach proposed in this paper over the frame-to-frame approach, such as PointMBF. Leveraging our neural implicit field constructed on RGB-D sequences, our approach excels in handling multi-view inconsistency. Consequently, the rerendering of neural implicit field can effectively leverage both photometric and geometric consistency to optimize the estimated pose, surpassing the capabilities of frame-to-frame methods.

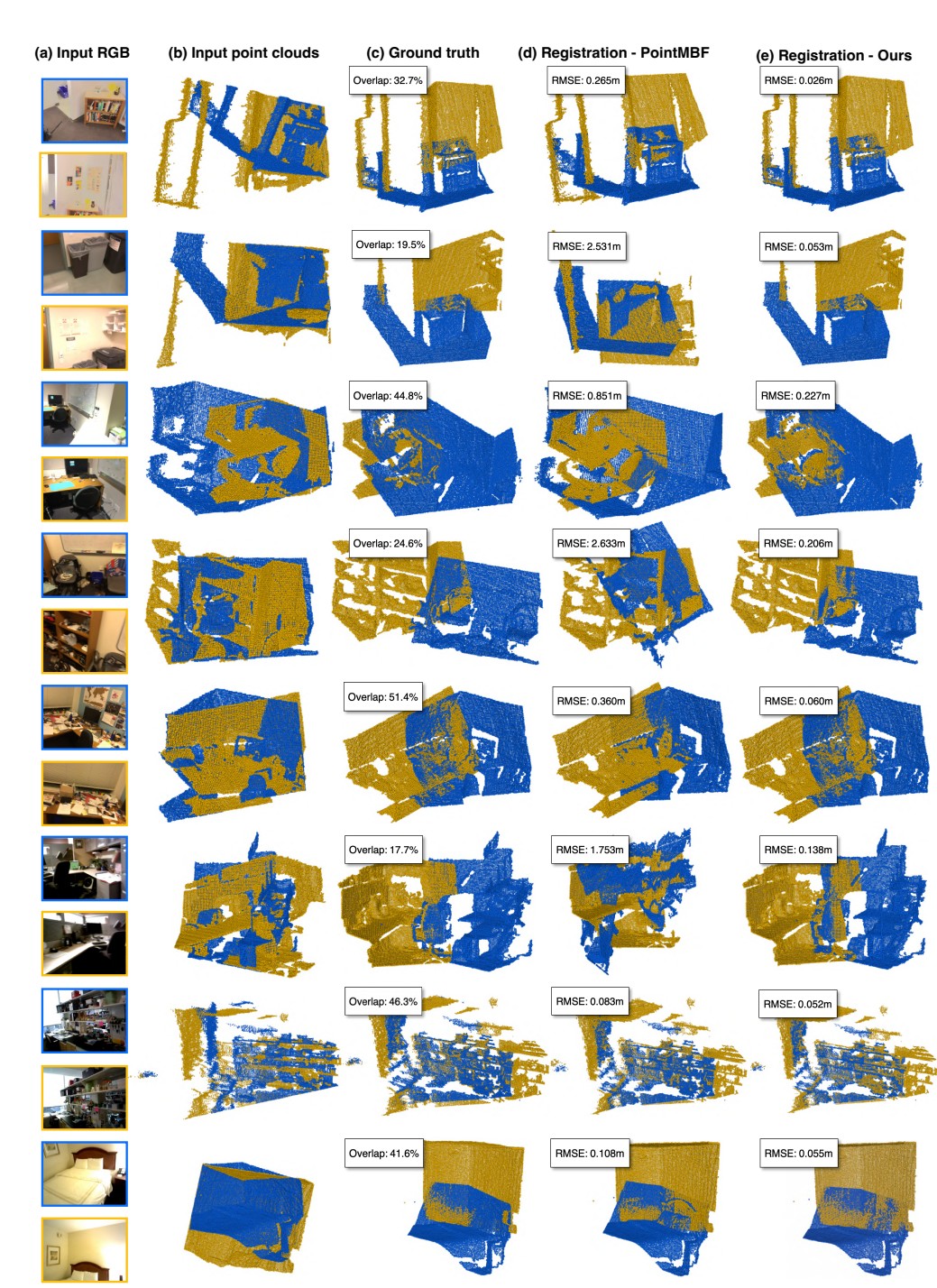

Figure 7: **Visual Comparison between PointMBF and F2M-Reg.** The registration model of PointMBF is trained on the ScanNet dataset Dai et al. (2017). Similarly, our registration model is fine-tuned on the ScanNet dataset, consistent with the experiments detailed in the paper.

