# OpenReview forum: "F2M-Reg: Unsupervised RGB-D Registration with Frame-to-Model Optimization"
_ICLR.cc/2025/Conference — Submitted to ICLR 2025_

### Official Review · Reviewer_bx3G · 2024-11-02

**Soundness:** 3
**Presentation:** 3
**Contribution:** 3
**Rating:** 6
**Confidence:** 4

**Summary:**

The paper introduced a method for RGB-D registration using frame-to-model optimization. The pipeline incorporates a pretrained registration model suing PointMBF and Co-SLAM for point cloud registration. The experiments on real-world ScanNet and 3DMatch datasets present the superiority of the proposed method. The ablation studies investigated the effectiveness of each components.

**Strengths:**

- The writing is comprehensive and easy to follow
- The proposed method demonstrates high performance compared to baseline methods on two real-world RGB-D datasets.
- The ablation study on the main text and supplementary materials are thorough and effectively showcases the efficacy of the proposed framework.

**Weaknesses:**

- The proposed method should not be characterized as unsupervised learning as it claimed. Instead, it involves a registration model that has been pretrained on a synthetic RGB-D dataset and subsequently adapted to real-world RGB-D scenes during inference. Strictly speaking, the method aligns more precisely with a zero-shot learning approach. Please ensure the claims are accurately represented.

- The experiments could benefit from utilizing more recent datasets. The ScanNet-v2 dataset, while widely used, is dated and known for sensor noise that results in unreliable depth maps with many gaps containing zero or infinity values. More accurate and high-resolution RGB-D streams are available in newer datasets like ScanNet++ and TUM RGB-D. Conducting additional experiments on these recent datasets is recommended.

- The proposed pipeline incorporates Co-SLAM, which utilizes a tracking system for camera pose estimation, requiring that the input RGB-D stream strictly follow a time series. However, this requirement shows limitations for general RGB-D registration tasks that handle multi-view data, where the input does not necessarily adhere to a time-series format. In scenarios involving unordered data, the effectiveness of a SLAM system may be compromised.

**Questions:**

- The paper states that the frame-to-frame framework experiences difficulties in maintaining multi-view consistency due to issues like lighting variations, geometric occlusions, and reflective surfaces; however, NeRF encounters similar challenges under these conditions. It should be clarified how the frame-to-model approach addresses these limitations in comparison to the frame-to-frame method.

- In the proposed framework, initial poses are generated using a bootstrap method trained on synthetic RGB-D data. Considering that Co-SLAM, utilized for tracking within the proposed framework, also incorporates its pose initialization strategy based on constant movement assumption, how do these initial poses from the bootstrap method align or integrate with the initial pose assumptions in Co-SLAM? Specifically, please detail any adaptations or refinements made to ensure consistency between the poses initialized by F2M-Reg and the subsequent pose tracking performed by Co-SLAM,

- Please also discuss the limitations of the proposed methods.

**Details Of Ethics Concerns:**

No ethics concern.

---

> ### Author Response · Authors · 2024-11-23
>
> **Q1: The claim about 'unsupervised RGB-D Registration' / 'unsupervised learning'**
>
> Following previous work [1] [2] [3] [4], unsupervised RGB-D registration aims to training the registration model without the supervision of the ground-truth poses. This line of work usually first computes the pseudo poses between the point cloud pairs, and trains the model based on the pseudo poses. Our method also follows this paradigm, with a novel frame-to-model optimization method to obtain accurate pseudo poses. And we propose a model initialization method based on synthetic data. On the other hand, zero-shot learning focuses on predicting novel classes or domains without the corresponding data, which is different from this task.
>
> **Q2: More evaluations on different datasets.**
>
> We compare our method with PointMBF on TUM RGB-D under the $50$-frame setting. And our method surpasses PointMBF by $9.4$ pp on Rotation Accuracy@$5^{\circ}$, $12$ pp on Translation Accuracy@$5\text{cm}$, and $9.5$ pp on Chamfer Accuracy@$1\text{cm}$. These results have proven the strong generality of method to new datasets. We will add these results in the future version.
>
> |               | Rot(5) | Rot(10) | Rot(45) | Trans(5) | Trans(10) | Trans(45) | Chamfer(1) | Chamfer(5) | Chamfer(10) |
> | :-----------: | :----: | :-----: | :-----: | :------: | :-------: | :-------: | :--------: | :---------: | :---------: |
> |   PointMBF    |  85.9  |  97.9   |  100.0  |   66.5   |   84.3    |   98.4    |    69.6    |    86.4     |    91.6     |
> | F2M-Reg(full) |  95.3  |  96.9   |  100.0  |   78.0   |   94.8    |   99.5    |    79.1    |    95.3     |    96.9     |
>
> **Q3:   Work on the unordered data.**
>
> It is common practice for unsupervised RGB-D registration methods [1] [2] [3] [4] to train the registraion model with RGB-D sequences. The main reason here is to guarantee that there is reasonable overlap between two frames, which is difficult to achieve with unordered data. It is the overlap that counts, rather than the orderness. And this is the same for Co-SLAM part. Theoretically, if the unordered data are organized to guarantee the overlap between frames, our method should also work. Besides, our method only leverages RGB-D sequences during training, and we do not rely on RGB-D sequences during inference and thus can handle various multi-view RGB-D data in applications.
>
> **Q4:  The reason on why F2M approach superior to F2F approach.**
>
> Neural implicit field still has challenges in handling multi-view inconsistency, especially in extreme cases. However, compared to the traditional rasterization process used in the frame-to-frame methods, neural implicit field is well-known for its stronger capability to handle multi-view inconsistency as noted in NeRF [5], NeuS [6], etc. By encoding the view direction information, the neural implicit field can render more geometrically and photometrically consistent images, which helps optimize a more accurate pose to train the registration model. On the contrary, the frame-to-frame method only project one frame into the other and compare the consistency between two images, which is more easily affected by the multi-view inconsistency.

---

> > ### Author Response · Authors · 2024-11-23
> >
> > **Q5:   Detail on the initial poses in the fine-tuning stage.**
> >
> > We omit the original constant movement-based pose initialization in Co-SLAM and directly use the pose predicted by our model as the initial pose. In this task, the distance between two frames are more distant than a traditional SLAM task, i.e., the frame pairs are $20$ frames apart in our case, so the constant movement assumption does not hold anymore. And we then optimize the predicted pose in the tracking and the mapping stages, which are used to train the registration model.
> >
> > **Q6:  Limitation.**
> >
> > In spite of the state-of-the-art performance, there are still some limitations in our method. (1) Our method cannot be directly used in outdoor scenes. On the one hand, the excessive foreground and background depth variations can cause the subscene to be incorrectly constructed. On the other hand, it is difficult to encode a large open scene with one neural implicit field. A possible solution to this problem is to leverage multiple local neural fields like block-NeRF. We will leave this as a future work. (2) Our method requires relatively long training time due to the optimization of the neural field. We can reduce the tracking iterations to balance the accuracy and the efficiency. And leveraging more efficient neural representations such as 3D Gaussian is also a promising research direction. We will add the discussions in the future version.
> >
> > [1]: Mohamed El Banani, Luya Gao, and Justin Johnson. Unsupervisedr&r: Unsupervised point cloud registration via differentiable rendering. CVPR 2021, pp. 7129-7139.
> >
> > [2]: Mohamed El Banani and Justin Johnson. Bootstrap your own correspondences. ICCV 2021, pp. 6433-6442.
> >
> > [3]: Ziming Wang, Xiaoliang Huo, Zhenghao Chen, Jing Zhang, Lu Sheng, and Dong Xu. Improving rgb-d point cloud registration by learning multi-scale local linear transformation. ECCV 2022, pp. 175-191.
> >
> > [4]: Mingzhi Yuan, Kexue Fu, Zhihao Li, Yucong Meng, and Manning Wang. Pointmbf: A multi-scale bidirectional fusion network for unsupervised rgb-d point cloud registration. ICCV 2023, pp. 17694-17705.
> >
> > [5]: Mildenhall B, Srinivasan P P, Tancik M, et al. Nerf: Representing scenes as neural radiance fields for view synthesis[J]. Communications of the ACM, 2021, 65(1): 99-106.
> >
> > [6]: Wang P, Liu L, Liu Y, et al. Neus: Learning neural implicit surfaces by volume rendering for multi-view reconstruction[J]. arXiv preprint arXiv:2106.10689, 2021.

---

> ### Comment · Reviewer_bx3G · 2024-11-27
> **Response to Rebuttal**
>
> I really appreciate the comprehensive responses from the authors. Some of my concerns regarding the F2F and F2M, the integration of Co-SLAM, and the training strategies are sufficiently addressed. The authors provide additional experiments on TUM-RGBD and showing their superiority over the baseline method. However, no experiments on the recommended ScanNet++ (Yeshwanth et al.) are provided. Considering the performance issues also highlighted in other reviews, I keep my score unchanged.
>
> A minor suggestion: The PDF file can be modified and reuploaded to reflect the revisions. Instead of promising future changes, it is more straightforward to show and highlight the revisions directly.
>
>
> Yeshwanth, Chandan, et al. "Scannet++: A high-fidelity dataset of 3d indoor scenes." Proceedings of the IEEE/CVF International Conference on Computer Vision. 2023.

---

> > ### Author Response · Authors · 2024-12-03
> >
> > **Q1: Evaluation on ScanNet++.**
> >
> > **A:** The table below presents the results of our tests on ScanNet++ iPhone RGB-D sequences under the 20-frame setting. Our method outperforms PointMBF significantly, with a $12.0$ pp improvement in Rotation Accuracy@$5^{\circ}$, a $16.6$ pp increase in Translation Accuracy@$5 cm$, and a $14.2$ pp boost in Chamfer Accuracy@$1 cm$. These results, evaluated on the ScanNet++ dataset, further demonstrate the strong generality of our method to new datasets.
> >
> > |          | Rot(5) | Rot(10) | Rot(45) | Trans(5) | Trans(10) | Trans(45) | Chamfer(5) | Chamfer(10) | Chamfer(45) |
> > | :------: | :----: | :-----: | :-----: | :------: | :-------: | :-------: | :--------: | :---------: | :---------: |
> > | F2M-Reg  |  95.7  |  98.2   |  99.4   |   79.7   |   92.9    |   97.7    |    88.3    |    96.9     |    98.0     |
> > | PointMBF |  83.7  |  90.3   |  96.5   |   63.1   |   78.0    |   89.2    |    74.1    |    86.8     |    89.9     |
> >
> > **Q2: The performance issue.**
> >
> > **A:** The performance issues raised by other reviewers stem from comparisons between the PointMBF and the F2M approach without warm-up. As emphasized in the paper, our frame-to-model approach is sensitive to the initialization of the registration model, which is why we designed the synthetic warm-up mechanism to mitigate this challenge. To further highlight this, we compared the F2F and F2M approaches with synthetic warm-up in **Tab. 11** of the main paper, which clearly demonstrates the superiority of our F2M approach in achieving robust and accurate registration performance.

---

> > > ### Comment · Reviewer_bx3G · 2024-12-03
> > > **Which scenes were evaluated in ScanNet++**
> > >
> > > Thanks for the additional experiments, I would like to ask which scenes are evaluated on ScanNet++ dataset. I assume the presented results are averaged outcomes on several scenes.

---

> > > > ### Author Response · Authors · 2024-12-03
> > > >
> > > > Yes, the results presented are average values across multiple scenes. Due to the large size of the dataset, we evaluated only 74 scenes (380 in total) from the iPhone RGB-D sequences in the table above. We will include results on both the DSLR and iPhone RGB-D sequences in the future version.

---

> ### Comment · Reviewer_bx3G · 2024-12-03
> **Response to additional experiments**
>
> The additional experiments present a solid demonstration for the effectiveness of the proposed method. I suggest incorporating both the TUM-RGBD and ScanNet++ results into the revised paper. Considering the thorough response, **I am pleased to raise my score to 6 (or 7 if that were an option) to reflect the good quality of the paper.**

---

> > ### Author Response · Authors · 2024-12-03
> >
> > Thanks for your valuable feedbacks and recognition of our work!

---

### Official Review · Reviewer_Qbae · 2024-11-02

**Soundness:** 3
**Presentation:** 3
**Contribution:** 3
**Rating:** 8
**Confidence:** 4

**Summary:**

The paper introduces an unsupervised method for robust RGB-D registration without ground-truth pose supervision. Unlike prior methods that rely on frame-to-frame photometric and geometric consistency, which are often affected by lighting changes, occlusion, and reflective materials, F2M-Reg employs a frame-to-model approach. The method begins with pre-training the model on a synthetic dataset, Sim-RGBD, with ground-truth poses, and subsequently fine-tunes it on real-world datasets without ground-truth poses by leveraging a neural implicit field as a 3D scene representation for pose optimization. This approach enhances robustness against multi-view inconsistencies, as demonstrated by experimental results comparing F2M-Reg with existing methods. In summary, F2M-Reg contributes a new unsupervised RGB-D registration framework, a synthetic dataset for initial model training, and an effective frame-to-model approach, setting new benchmarks on popular RGB-D datasets.

**Strengths:**

The paper identifies the limitations of frame-to-frame matching in unsupervised learning, particularly due to instabilities arising from lighting variations, occlusions, and reflective surfaces. The proposed frame-to-model matching, supported by a neural implicit field (NeRF), effectively mitigates these issues, demonstrating a significant improvement over traditional methods.

**Weaknesses:**

- The use of the term bootstrap in the paper is potentially misleading. In deep learning, bootstrapping generally refers to iterative self-training, where a model refines itself by generating pseudo-labels and learning from them. The training on synthetic data described in this paper aligns more with pre-training. However, the fine-tuning on real datasets with initial poses refined through NeRF could be called bootstrapping.

**Questions:**

- The performance gains from Sim-RGBD bootstrapping across different datasets is not quite consistent. Can the authors provide insights into why bootstrapping appears less critical for ScanNet compared to 3DMatch?

- NeRF optimization, particularly joint pose and neural field optimization, typically assumes static scenes. This assumption might limit the method's performance in dynamic environments. If this is indeed a constraint, it should be acknowledged in the paper. If not, could the authors clarify why the method remains effective in dynamic scenarios?

---

> ### Author Response · Authors · 2024-11-23
>
> **Q1: About the definition of 'bootstrap'.**
>
> We use the term 'bootstrap' in this paper to describe the process of initializing the model for more effective unsupervised training. This process is achieved by leveraging the synthetic data. We admit that the usage of this term is somewhat inaccurate, and we will replace this term in the future version.
>
> **Q2: Synthetic Bootstrapping achieves more improvements on 3DMatch than ScanNet.**
>
> The reasons for this inconsistency are two-fold. First, as we use ScanNet for testing in all the experiments, there is naturally a domain gap for the model trained on 3DMatch which limits its performance. Second, ScanNet contains more scenes than 3DMatch ($1045$ v.s. $71$ for training), which also affects the generalization of the model. By leveraging the pre-training, the various synthetic data and the supervised training manner helps fill this gap and improve the generalization of the model.
>
> **Q3: Effectiveness in dynamic scenarios.**
>
> Yes, this paper focuses on the unsupervised RGB-D registration For the pre-training stages,in static scenarios like previous work UR&R [1], BYOC [2], LLT [3], PointMBF [4] and unsupervised RGB-D registration in dynamic scenarios is out of the scope of this paper. However, we would note that there are also neural implicit fields for dynamic scenes, which could be integrated into our pipeline to solve the dynamic problem. We will leave this as the future work.
>
> [1]: Mohamed El Banani, Luya Gao, and Justin Johnson. Unsupervisedr&r: Unsupervised point cloud registration via differentiable rendering. CVPR 2021, pp. 7129-7139.
>
> [2]: Mohamed El Banani and Justin Johnson. Bootstrap your own correspondences. ICCV 2021, pp. 6433-6442.
>
> [3]: Ziming Wang, Xiaoliang Huo, Zhenghao Chen, Jing Zhang, Lu Sheng, and Dong Xu. Improving rgb-d point cloud registration by learning multi-scale local linear transformation. ECCV 2022, pp. 175-191.
>
> [4]: Mingzhi Yuan, Kexue Fu, Zhihao Li, Yucong Meng, and Manning Wang. Pointmbf: A multi-scale bidirectional fusion network for unsupervised rgb-d point cloud registration. ICCV 2023, pp. 17694-17705.

---

> > ### Comment · Reviewer_Qbae · 2024-11-28
> >
> > Thanks for the response. It has addressed my concerns. I've upgraded my rating to accept and change presentation from fair to good to reflect this.

---

> ### Author Response · Authors · 2024-11-29
>
> Thank you very much for recognizing our work and for the honor of us being able to address your concerns!

---

### Official Review · Reviewer_Sxy2 · 2024-11-03

**Soundness:** 2
**Presentation:** 3
**Contribution:** 2
**Rating:** 5
**Confidence:** 2

**Summary:**

This paper presents F2M-Reg, an unsupervised RGB-D registration framework that addresses the frame-to-frame registration task by dealing with multi-view inconsistencies with bootstrapping with a synthetic dataset.

**Strengths:**

1. F2M-Reg stands out by shifting from a frame-to-frame to a frame-to-model approach for RGB-D registration, which is an extension of existing approaches in the context of unsupervised 3D vision tasks.
2. This use of a neural implicit field as a global scene model to capture broader scene-level information is a possible direction to handle complex conditions, such as low overlap and lighting changes, where traditional methods often fall short.
 3. The introduction of a synthetic bootstrapping dataset, Sim-RGBD, bridges the gap between synthetic and real-world performance in unsupervised settings, which is a notable improvement in unsupervised model initialization.

**Weaknesses:**

1. Although F2M-Reg is compared with several baselines, it is unclear where the improvement comes from. According to Table 4, the results without bootstrapping are not exciting enough.
2. In order to evaluate the effectiveness of the neural implicit field-guided mechanism, this paper needs additional experiments and comparisons with SOTA approaches without bootstrapping.

**Questions:**

See weakness.

---

> ### Author Response · Authors · 2024-11-23
>
> **Q1:  Where the improvements come from.**
>
> We have separately studied the improvements from each component in Tab.2, Tab.3, Tab.4 and Tab.5 in the main paper. Here, we reorganize the results in the table below for a better understanding. BS refers to the synthetic bootstrap, F2M refers to frame-to-model optimization, and F2F refers to the frame-to-frame optimization. Independently applying synthetic bootstrap and frame-to-model optimization both achieves significant improvements compared with the frame-to-frame baseline, demonstrating the strong effectiveness of our designs. And applying both of them further improves the results, achieving the improvements of $6$ pp over the BS-only model and $3$ pp over the F2M-only model on most metrics. We will refine the descriptions in the future version.
>
> | BS           |     F2F      |     F2M      | Rot(5) | Rot(10) | Rot(45) | Trans(5) | Trans(10) | Trans(45) | Chamfer(1) | Chamfer(5) | Chamfer(10) |
> | ------------ | :----------: | :----------: | :----: | :-----: | :-----: | :------: | :-------: | :-------: | :--------: | :---------: | :---------: |
> |              | $\checkmark$ |              |  60.4  |  68.2   |  79.9   |   40.0   |   54.3    |   66.9    |    48.9    |    61.5     |    65.8     |
> | $\checkmark$ |              |              |  71.3  |  78.6   |  87.4   |   46.9   |   65.5    |   76.3    |    57.5    |    72.2     |    75.0     |
> |              |              | $\checkmark$ |  74.4  |  82.8   |  92.3   |   46.8   |   67.9    |   80.4    |    58.5    |    75.5     |    79.0     |
> | $\checkmark$ | $\checkmark$ |              |  75.2  |  82.5   |  90.3   |   47.4   |   68.3    |   80.5    |    58.8    |    76.5     |    79.4     |
> | $\checkmark$ |              | $\checkmark$ |  77.4  |  84.5   |  92.5   |   50.0   |   70.6    |   82.1    |    61.5    |    77.6     |    80.9     |
>
>
> **Q2:   The effectiveness of the neural implicit field-guided mechanism.**
>
> The results have already been shown in the Tab.2 and Tab.4 of the main paper and we put them together in the table below. Under the $50$-frame setting, our method without bootstrap surpasses the previous state-of-the-art method PointMBF by $14$ pp on Rotation Accuracy@$5^{\circ}$, $6.8$ pp on Translation Accuracy@$5\text{cm}$ and $9.6$ pp on Chamfer Distance Accuracy@$5\text{cm}$, showing strong superiority.
>
> |                        | Rot(5) | Rot(10) | Rot(45) | Trans(5) | Trans(10) | Trans(45) | Chamfer(1) | Chamfer(5) | Chamfer(10) |
> | :--------------------: | :----: | :-----: | :-----: | :------: | :-------: | :-------: | :--------: | :---------: | :---------: |
> |        PointMBF        |  60.4  |  68.2   |  79.9   |   40.0   |   54.3    |   66.9    |    48.9    |    61.5     |    65.8     |
> | F2M-Reg(w/o bootstrap) |  74.4  |  82.8   |  92.3   |   46.8   |   67.9    |   80.4    |    58.5    |    75.5     |    79.0     |
> |     F2M-Reg(full)      |  77.4  |  84.5   |  92.5   |   50.0   |   70.6    |   82.1    |    61.5    |    77.6     |    80.9     |
>
> We further show the comparisons under the $20$-frame setting in the table below. As this setting is relatively easy as the frame pairs share larger overlap, our method without bootstrap still outperforms the PointMBF baseline.
>
> |                        | Rot(5) | Rot(10) | Rot(45) | Trans(5) | Trans(10) | Trans(45) | Chamfer(1) | Chamfer(5) | Chamfer(10) |
> | :--------------------: | :----: | :-----: | :-----: | :------: | :-------: | :-------: | :--------: | :---------: | :---------: |
> |        PointMBF        |  96.0  |  97.6   |  98.9   |   83.9   |   93.8    |   97.7    |    92.8    |    97.3     |    97.9     |
> | F2M-Reg(w/o bootstrap) |  96.8  |  98.9   |  99.8   |   83.6   |   95.2    |   98.7    |    92.1    |    98.2     |    98.8     |
> |     F2M-Reg(full)      |  97.6  |  99.1   |  99.8   |   85.5   |   95.8    |   98.8    |    93.1    |    98.4     |    98.9     |

---

> > ### Comment · Reviewer_Sxy2 · 2024-11-26
> > **Seems only improving on the choosen ScanNet dataset**
> >
> > Thanks for the author's response, but it still does not fully address my concern:
> > 1. The summarized improvement on the 50-frame setting only works on the ScanNet dataset, but according to Tab. 2 and Tab. 4, the performance of F2M on 3DMatch is lower than PointMBF on Rotation 5 and Translation 5.
> > 2. On the more common 20-frame setting, the improvement of F2M w/o bootstrapping is marginal on the ScanNet dataset. The comparison on 3DMatch is not reported.
> > 3. PointMBF is not designed for low-overlap scenarios like 50 frames apart, but this is not clearly discussed in the paper.
> >
> > The differences between the two datasets should be discussed and analyzed explicitly.

---

> ### Author Response · Authors · 2024-11-29
>
> **Q1: About the performance of w/o warm-up registration model.**
>
> **A:** As already described in the paper, our frame-to-model method is more sensitive to model initialization as it needs to track the poses of more frames to optimize a neural implicit field, as we design a synthetic bootstrap method to solve this problem. To further investigate the effectiveness of our approach, we compared the results of applying synthetic warm-up to the frame-to-frame method, PointMBF. The results are shown in the tables below, where both methods were trained on 3DMatch and ScanNet. The first table presents the comparison under the 50-frame-apart setting, while the second table corresponds to the 20-frame-apart setting. Across both settings and training datasets, our method consistently outperforms the frame-to-frame baseline, even with synthetic warm-up applied.
>
> |                   | Train Set | Rot(5) | Rot(10) | Rot(45) | Trans(5) | Trans(10) | Trans(45) | Chamfer(5) | Chamfer(10) | Chamfer(45) |
> | :---------------: | :-------: | :----: | :-----: | :-----: | :------: | :-------: | :-------: | :--------: | :---------: | :---------: |
> | F2F(with warm-up) |  3DMatch  |  71.3  |  79.2   |  87.7   |   43.9   |   64.6    |   77.0    |    55.5    |    72.7     |    76.0     |
> | F2M(with warm-up) |  3DMatch  |  72.6  |  81.1   |  91.1   |   44.6   |   65.9    |   78.5    |    56.1    |    73.6     |    77.1     |
> | F2F(with warm-up) |  ScanNet  |  75.2  |  82.5   |  90.3   |   47.4   |   68.3    |   80.5    |    58.8    |    76.5     |    79.4     |
> | F2M(with warm-up) |  ScanNet  |  77.4  |  84.5   |  92.5   |   50.0   |   70.6    |   82.1    |    61.5    |    77.6     |    80.9     |
>
> |                   | Train Set | Rot(5) | Rot(10) | Rot(45) | Trans(5) | Trans(10) | Trans(45) | Chamfer(5) | Chamfer(10) | Chamfer(45) |
> | :---------------: | :-------: | :----: | :-----: | :-----: | :------: | :-------: | :-------: | :--------: | :---------: | :---------: |
> | F2F(with warm-up) |  3DMatch  |  96.1  |  98.5   |  99.6   |   81.7   |   94.5    |   98.4    |    91.2    |    97.7     |    98.5     |
> | F2M(with warm-up) |  3DMatch  |  96.3  |  98.7   |  99.7   |   81.9   |   94.7    |   98.5    |    91.4    |    97.9     |    98.6     |
> | F2F(with warm-up) |  ScanNet  |  97.3  |  99.0   |  99.7   |   83.7   |   95.5    |   98.7    |    92.5    |    98.3     |    98.8     |
> | F2M(with warm-up) |  ScanNet  |  97.6  |  99.1   |  99.8   |   85.5   |   95.8    |   98.8    |    93.1    |    98.4     |    98.9     |
>
> It is also worth mentioning that our method was trained for only 1 epochs on 3DMatch, whereas PointMBF was trained for 12 epochs. To provide a clearer comparison, we report the results after training for 4 epochs without bootstrapping in the following table. As shown, our method continues to demonstrate improvements, further validating its effectiveness.
>
> |                | Rot(5) | Rot(10) | Rot(45) | Trans(5) | Trans(10) | Trans(45) | Chamfer(5) | Chamfer(10) | Chamfer(45) |
> | :------------: | :----: | :-----: | :-----: | :------: | :-------: | :-------: | :--------: | :---------: | :---------: |
> |    PointMBF    |  59.3  |  62.5   |  76.6   |   34.2   |   47.9    |   61.6    |    42.9    |    55.8     |    60.2     |
> | F2M (1 epoch)  |  59.0  |  71.3   |  86.3   |   30.4   |   52.3    |   68.0    |    43.2    |    62.1     |    66.5     |
> | F2M (4 epochs) |  66.3  |  77.5   |  90.2   |   36.5   |   58.7    |   74.8    |    48.5    |    68.7     |    73.4     |

---

> ### Author Response · Authors · 2024-11-29
>
> **Q2: About the results under the 20-frame setting.**
>
> **A:** The $20$-frame setting is more like a *easy* setting instead of a *common* one as the viewpoint changes are small, and thus the performance tends to be saturated. However, in real applications, the $20$-frame setting does not always hold. For example, in the scenes with fast motion or low frame rate, the frame-to-frame methods could fail but our method still provides accurate pose estimations.
>
> And further we further report the results on 3DMatch in the table below.
>
> |                        | Rot(5) | Rot(10) | Rot(45) | Trans(5) | Trans(10) | Trans(45) | Chamfer(5) | Chamfer(10) | Chamfer(45) |
> | :--------------------: | :----: | :-----: | :-----: | :------: | :-------: | :-------: | :--------: | :---------: | :---------: |
> | PointMBF(w/o warm-up)  |  94.6  |  97.0   |  98.7   |   81.0   |   92.0    |   97.1    |    91.3    |    96.6     |    97.4     |
> |   F2M (w/o warm-up)    |  95.7  |  98.6   |  99.7   |   79.8   |   94.2    |   98.5    |    90.5    |    97.8     |    98.5     |
> | PointMBF(with warm-up) |  96.1  |  98.5   |  99.6   |   81.7   |   94.5    |   98.4    |    91.2    |    97.7     |    98.5     |
> |   F2M (with warm-up)   |  96.3  |  98.7   |  99.7   |   81.9   |   94.7    |   98.5    |    91.4    |    97.9     |    98.6     |
>
> **Q3: About the low-overlap cases.**
>
> **A:** First of all, we would note that our method and PointMBF share the similar registration model, but differ in the training methods, i.e., frame-to-model v.s. frame-to-frame. It is one of the most disadvantages of the frame-to-frame methods that they cannot estimate accurate poses for the low-overlap pairs during training, which leads to suboptimal convergence of the model. On the contrary, our frame-to-model can provide better poses for the low-overlap pairs, which contributes to a more robust model, as shown in Tab.2 of the main paper. These results explicitly demonstrate the superiority of our method. Morever, we compare the poses during training estimated by both methods on pairs with rotation > $20^{\circ}$ on ScanNet scene0000_00 in the below table. It can be seen that our F2M method achieves obviously better pose optimization during training, providing the registration model with better supervised signals.
>
> |      | Rot(mean) | Rot(median) | Trans(mean) | Trans(median) |
> | :--: | :-------: | :---------: | :---------: | :-----------: |
> | F2F  |    4.9    |     2.2     |    13.0     |      6.0      |
> | F2M  |    2.9    |     0.8     |     7.0     |      3.0      |
>
> **Q4: About the differences between ScanNet and 3DMatch.**
>
> **A:** We show the statistics of the rotations, the translations, and the overlap ratios between the training pairs for both datasets. It can be seen that the two datasets do not differ much in data distribution. However, the size of the two datasets shows significant differences, i.e., 1045 vs 71, which could be the main cause of the performance differences.
>
> |         | Rot(mean) | Rot(median) | Trans(mean) | Trans(median) | Overlap(mean) | Overlap(median) |
> | :-----: | :-------: | :---------: | :---------: | :-----------: | :-----------: | :-------------: |
> | ScanNet |   11.9    |    11.0     |    18.2     |     16.0      |     63.6%     |       0.7       |
> | 3DMatch |   10.2    |     8.4     |    17.1     |     14.4      |     65.1%     |       0.7       |

---

### Official Review · Reviewer_4zeD · 2024-11-03

**Soundness:** 2
**Presentation:** 2
**Contribution:** 2
**Rating:** 5
**Confidence:** 4

**Summary:**

This paper proposes a frame-to-model optimization framework guided by a neural implicit field for unsupervised RGB-D registration. By introducing the differential rendering capabilities of neural radiance fields, better pose supervision can be achieved.  Meanwhile，this paper creates a synthetic dataset for warming up registration model.

**Strengths:**

1. The paper introduces neural radiance fields to provide global information for the registration training
2. It builds a synthetic dataset to pretrain the model, ensuring the rationality of pose initialization.

**Weaknesses:**

1. The comparison with existing work seems unfair:

   - sota methods like PointMBF are trained purely without supervision while the proposed method requires extra dataset with pose labels. In Table 1, the metrics of the proposed method looks slight better than existing unsupervised data. Does the improvement come from the benefit of including extra dataset? Please provide results without the boostrap using the process of training the registration model on the Sim-RGBD dataset for Table 1. Or please consider other ways to make a fair comparison. Will other work benefit the extra dataset with pose labels as well?

   -  In line 368-370, the authors propose to evaluate more difficult setting “evaluating view pairs sampled 50 frames apart” and mentioned “However, due to insufficient overlap in some segments of the data for pairs sampled 50 frames apart, the evaluation significantly distorts both the mean and median values. As a result, we have chosen not to include these results in our experimental presentation.” My question: why this 50 frames apart setting? Also if the insufficient overlap pairs are excluded,

2. The influence of the warming up dataset is not studied. How large the warming up dataset should be? How similar the warming up dataset should be to the target datasets? What’s is the influence of number objects of ShapeNet on the final metrics? For the construction of datasets, there are already many synthetic datasets for indoor scenes, such as Replica and Habitat. This paper does not demonstrate the advantages of the custom dataset compared to other datasets.

3. The training efficiency. As NeRF is very slow in training, the proposed method should require much more computation resources than PointMBF and hard to scale.

**Questions:**

See weakness

---

> ### Author Response · Authors · 2024-11-23
>
> **Q1:   The effectiveness of the warming up dataset.**
>
> |                        | Rot(5) | Rot(10) | Rot(45) | Trans(5) | Trans(10) | Trans(45) | Chamfer(1) | Chamfer(5) | Chamfer(10) |
> | :--------------------: | :----: | :-----: | :-----: | :------: | :-------: | :-------: | :--------: | :---------: | :---------: |
> |        PointMBF        |  96.0  |  97.6   |  98.9   |   83.9   |   93.8    |   97.7    |    92.8    |    97.3     |    97.9     |
> | F2M-Reg(w/o bootstrap) |  96.8  |  98.9   |  99.8   |   83.6   |   95.2    |   98.7    |    92.1    |    98.2     |    98.8     |
> |     F2M-Reg(full)      |  97.6  |  99.1   |  99.8   |   85.5   |   95.8    |   98.8    |    93.1    |    98.4     |    98.9     |
>
> In the table above, we compare PointMBF, F2M-Reg w/o bootstrapping, and full F2M-Reg under the $20$-frame setting on ScanNet. Our method still outperforms PointMBF even without bootstrapping, especially it can reduce extremely erroneous registrations. We further show the results under the $50$-frame setting (see the table below) with more severe multi-view inconsistency due to light variations and geometric occlusion. In this setting, our improvements are more significant, i.e., $14$ pp on Rotation Accuracy@$5^{\circ}$, $6.8$ pp on Translation Accuracy@$5\text{cm}$, and $9.6$ pp on Chamfer Distance Accuracy@$5\text{cm}$. These results further demonstrate the superiority of our frame-to-model design.
>
> |                        | Rot(5) | Rot(10) | Rot(45) | Trans(5) | Trans(10) | Trans(45) | Chamfer(1) | Chamfer(5) | Chamfer(10) |
> | :--------------------: | :----: | :-----: | :-----: | :------: | :-------: | :-------: | :--------: | :---------: | :---------: |
> |        PointMBF        |  60.4  |  68.2   |  79.9   |   40.0   |   54.3    |   66.9    |    48.9    |    61.5     |    65.8     |
> | F2M-Reg(w/o bootstrap) |  74.4  |  82.8   |  92.3   |   46.8   |   67.9    |   80.4    |    58.5    |    75.5     |    79.0     |
> |     F2M-Reg(full)      |  77.4  |  84.5   |  92.5   |   50.0   |   70.6    |   82.1    |    61.5    |    77.6     |    80.9     |
>
>
> **Q2:  50 frames apart in the evaluation phase.**
>
>
> The $50$-frame setting aims to evaluate the performance under low overlap, which is common in real applications with fast motion or low frame rate. It is more challenging due to the severe interference of the non-overlap region. In the evaluation, we exhaustively include all consecutive pairs which are $50$ frames apart, e.g., frame $1$ and $51$, frame $51$ and $101$, etc. We **do not exclude** any pairs even if they have extremely low overlap or no overlap. This further increase the difficulty of this setting. In the paper, we do not show the mean and the median errors in Tab.2 of the main paper, which we show below. The table below show the mean and median errors of the registration model trained on ScanNet. The results are clearly worse than those under the $20$-frame setting, but our method still outperforms PointMBF by a large margin.
>
> |               | Rotation Mean | Rotation Median | Translation Mean | Translation Median | Chamfer Mean | Chamfer Median |
> | :-----------: | :-----------: | :-------------: | :--------------: | :----------------: | :----------: | :------------: |
> |   PointMBF    |    19.1782    |     2.2901      |     38.0614      |       5.9361       |   85.7505    |     0.6877     |
> | F2M-Reg(full) |    15.5138    |     1.9196      |     30.0634      |       5.0210       |   73.8352    |     0.5405     |

---

> ### Author Response · Authors · 2024-11-23
>
> **Q3:  The influence of the warming up dataset.**
>
> **About the dataset size.** To investigate the influence of the dataset size, we use $20$, $40$, $60$ and $80$ scenes to bootstrap the model and directly evaluate without fine-tuning. The results are shown in the table below. It is observed that increasing the dataset size increases the performence, but the improvements are not significant. This means the synthetic bootstrap does not rely on a large synthetic dataset.
>
> |                     | Rot(5) | Rot(10) | Rot(45) | Trans(5) | Trans(10) | Trans(45) | Chamfer(1) | Chamfer(5) | Chamfer(10) |
> | :-----------------: | :----: | :-----: | :-----: | :------: | :-------: | :-------: | :--------: | :---------: | :---------: |
> | Sim_RGBD(20 scenes) |  72.7  |  79.8   |  88.4   |   47.9   |   66.7    |   77.3    |    58.4    |    73.4     |    76.3     |
> | Sim_RGBD(40 scenes) |  73.4  |  80.4   |  89.0   |   48.1   |   67.5    |   78.1    |    58.7    |    74.1     |    77.1     |
> | Sim_RGBD(60 scenes) |  72.6  |  79.5   |  88.2   |   48.1   |   66.5    |   77.1    |    58.4    |    73.2     |    76.1     |
> | Sim_RGBD(80 scenes) |  73.5  |  80.1   |  88.1   |   48.7   |   67.3    |   78.0    |    58.9    |    74.0     |    77.0     |
>
> **About the similarity.** Actually, our Sim-RGBD dataset is already very different from the real datasets. First, our dataset is constructed from ShapeNet, which contains the objects which cannot appear together in real scenes, such as planes and guitars, buses and chairs, etc. This induces inconsistent semantics and geometries between the synthetic and the real datasets. Second, our dataset puts the objects densely and randomly in the space, without collision detection or gravity, so there are severe layout differences between the synthetic and the real datasets. Nevertheless, the synthetic bootstrap still effectively provides a good initialization of the registration model, which demonstrates that the synthetic and the real datasets do not have to be very similar.
>
> **About other synthetic dataset.** We conducted an ablation study by training on the Replica dataset and testing on ScanNet. The results of these experiments are summarized in the table below. The findings indicate that training on Replica yields performance comparable to training on Sim_RGBD. This demonstrates the robustness and generality of our bootstrapping stage for utilizing simulation data across different datasets.
>
> |          | Rot(5) | Rot(10) | Rot(45) | Trans(5) | Trans(10) | Trans(45) | Chamfer(1) | Chamfer(5) | Chamfer(10) |
> | :------: | :----: | :-----: | :-----: | :------: | :-------: | :-------: | :--------: | :---------: | :---------: |
> | Replica  |  72.3  |  80.2   |  89.1   |   46.2   |   66.2    |   78.2    |    57.7    |    73.5     |    77.2     |
> | Sim_RGBD |  72.6  |  79.5   |  88.2   |   48.1   |   66.5    |   77.1    |    58.4    |    73.2     |    76.1     |
>
> **Q4:  Time efficiency.**
>
> Yes, our method increases the training time, mainly coming from the tracking and the mapping stages during optimizing the neural implicit field and the relative poses. However, we can reduce the tracking iterations (which we use $100$ in the experiments) for a balance between accuracy and efficiency. And as shown in Tab.7 of the main paper, reducing the tracking iterations only marginally influences the performance, but effectively reduces the training time.
> Moreover, we would also note that PointMBF uses $20\times$ training data than ours. PointMBF uses all pairs which are $20$ frames apart but we sample the training samples in a consecutive manner. For this reason, our method is more data-efficient than PointMBF.

---

> > ### Comment · Reviewer_4zeD · 2024-11-25
> > **If existing dataset performs better, why this synthetic dataset？and why this can be a contribtion?**
> >
> > Thanks authors for the experiments for datasets.
> >
> > I have two questions remains:
> >
> > 1) My question is about " What’s is the influence of number objects of ShapeNet on the final metrics?" while the results is about number of scenes. Could you please provide the number of the objects for the scenes used in the 20 -80 scenes?
> >
> > 2) My other concerns is if existing dataset performs better (according to results), why this synthetic dataset and why this can be a contribution? In which sense, this novel synthetic dataset fills the gap of these existing dataset？

---

> > > ### Author Response · Authors · 2024-11-29
> > >
> > > **Q1: Th influence of the number of the objects in the synthetic dataset.**
> > >
> > > **A:** Currently, there are $400$ objects per scene. We are currently working on re-rendering the dataset which has $50$ objects per scene and re-training the models. The results will be made available as soon as we finish.
> > >
> > > **Q2: The contribution of the synthetic dataset.**
> > >
> > > **A:** We would first note that our contribution lies in the warming-up mechanism using a synthetic dataset to mitigate the sensitivity of our frame-to-model optimization to the model initialization, rather than in the construction of the Sim-RGBD dataset itself. We construct our own dataset instead of using existing synthetic datasets such as Replica as they commonly contain a limited number of scenes, and it is more flexible and easy to build a large dataset at little cost by randomly placing objects in a space. And notably, warming-up with Sim-RGBD shows strong effectiveness in the experiments, which demonstrates the potential that such a simple synthetic dataset could benefit complicated 3D vision tasks, with little extra costs. And we hope this could inspire more future work on this topic.

---

> > > ### Author Response · Authors · 2024-12-03
> > >
> > > **Q1: The influence of the number of the objects in the synthetic dataset.**
> > >
> > > A: We conduct an additional ablation experiment on the number of objects in Sim_RGBD by rendering a dataset containing 50/400 objects per scene. The results, tested on ScanNet under the 50 frame apart setting with 20 scenes, are presented below. The sparse distribution of objects (Sim_RGBD with 50 objects per scene) leads to insufficient geometric information in the RGB-D data of each frame, limiting the training effectiveness for the registration model.
> > >
> > > |                    | Rot(5) | Rot(10) | Rot(45) | Trans(5) | Trans(10) | Trans(45) | Chamfer(5) | Chamfer(10) | Chamfer(45) |
> > > | :----------------: | :----: | :-----: | :-----: | :------: | :-------: | :-------: | :--------: | :---------: | :---------: |
> > > | Sim_RGBD(400 objs) |  72.7  |  79.8   |  88.4   |   47.9   |   66.7    |   77.3    |    58.4    |    73.4     |    76.3     |
> > > | Sim_RGBD(50 objs)  |  59.9  |  73.2   |  88.2   |   29.5   |   50.8    |   70.6    |    41.1    |    62.9     |    69.3     |

---

> ### Comment · Reviewer_4zeD · 2024-11-25
> **Seems no improvement in strict metrics when compared PointMBF**
>
> Thanks the authors for providing these extra results.
>
> From the table, we can conclude that the improvement on ScanNet of 20 frames over PointMBF mostly comes from the extra training data.  From the table, I may not agree with the conclusion that "Our method still outperforms PointMBF even without bootstrapping, especially it can reduce extremely erroneous registrations." In some metrics, Trans(5) Chamfer(5), the method is worse. I can say the two methods are about the same performance.
>
> From Table2 and  Table4  on 3DMatch of 50 frames apart setting, it also shows without introducing extra training data, the performance of the method measured by strict metrics standard is also similar (or worse?) to PointMBF. In the setting of 20frames in Table 1, will the result of the method with introducing extra dataset also similar to PointMBF or worse？
>                     Rot5 Tran5 Chamfer 1
> PointMBF    59.3  34.2 42.9
> The method 59.0  30.4  43.2
>
> Yet, the method has shown improvements on erroneous cases. Among all the comparison on different datasets, the method shows improvements on ScanNet on 50 frames apart and as authors said in the setting of 20 frames, the method also shows improvements over pointMBF on less strict metrics like large error threshold (Rot10Trans10) .

---

> ### Author Response · Authors · 2024-11-29
>
> **Q3: About the performance of w/o warm-up registration model.**
>
> **A:** As already described in the paper, the frame-to-model approach is more sensitive to the model initialization as it needs to track the poses of more frames to optimize a neural implicit field. This sensitivity prompted the design of our synthetic warm-up method to address the issue. For more comparisons, we apply the synthetic warming-up on both F2F and F2M approaches on both datasets in the following tables: the first table corresponds to the 50-frame setting, and the second table to the 20-frame setting. These comparisons highlight the superiority of the F2M approach over the F2F approach, demonstrating its effectiveness in training the registration model.
>
> |                   | Train Set | Rot(5) | Rot(10) | Rot(45) | Trans(5) | Trans(10) | Trans(45) | Chamfer(5) | Chamfer(10) | Chamfer(45) |
> | :---------------: | :-------: | :----: | :-----: | :-----: | :------: | :-------: | :-------: | :--------: | :---------: | :---------: |
> | F2F(with warm-up) |  3DMatch  |  71.3  |  79.2   |  87.7   |   43.9   |   64.6    |   77.0    |    55.5    |    72.7     |    76.0     |
> | F2M(with warm-up) |  3DMatch  |  72.6  |  81.1   |  91.1   |   44.6   |   65.9    |   78.5    |    56.1    |    73.6     |    77.1     |
> | F2F(with warm-up) |  ScanNet  |  75.2  |  82.5   |  90.3   |   47.4   |   68.3    |   80.5    |    58.8    |    76.5     |    79.4     |
> | F2M(with warm-up) |  ScanNet  |  77.4  |  84.5   |  92.5   |   50.0   |   70.6    |   82.1    |    61.5    |    77.6     |    80.9     |
>
> |                   | Train Set | Rot(5) | Rot(10) | Rot(45) | Trans(5) | Trans(10) | Trans(45) | Chamfer(5) | Chamfer(10) | Chamfer(45) |
> | :---------------: | :-------: | :----: | :-----: | :-----: | :------: | :-------: | :-------: | :--------: | :---------: | :---------: |
> | F2F(with warm-up) |  3DMatch  |  96.1  |  98.5   |  99.6   |   81.7   |   94.5    |   98.4    |    91.2    |    97.7     |    98.5     |
> | F2M(with warm-up) |  3DMatch  |  96.3  |  98.7   |  99.7   |   81.9   |   94.7    |   98.5    |    91.4    |    97.9     |    98.6     |
> | F2F(with warm-up) |  ScanNet  |  97.3  |  99.0   |  99.7   |   83.7   |   95.5    |   98.7    |    92.5    |    98.3     |    98.8     |
> | F2M(with warm-up) |  ScanNet  |  97.6  |  99.1   |  99.8   |   85.5   |   95.8    |   98.8    |    93.1    |    98.4     |    98.9     |

---

### Author Response · Authors · 2024-11-23

We thank all reviewer for their valuable comments, recognizing the interestring problem being solved(Reviewer Sxy2), the technical soundness of the method(All Reviewers), the good writing quality (Reviewer 4zeD), and intuitive illustration (Reviewer Sxy2, Reviewer Qbae), the comprehensive experiments (All Reviewers), the outstanding performance (All Reviewers), and the application value(Reviewer 4zeD). We will address each reviewer's concerns individually below.

---

### Author Response · Authors · 2024-11-27
**Summary of changes on rebuttal revision**

Dear reviewers,
We have made several modifications based on the your feedbacks and re-uploaded the PDF. The changes are summarized as follows:
- For **Reviewer 4zeD**, we updated all tables in the paper to report both the mean and median values of the registration models.
- For **Reviewer Sxy2**, we added an **Ablation on Different Module Combinations** in the Appendix to emphasize the paper's core contribution. We demonstrate that the F2M approach enables more effective unsupervised training of the registration model, with a synthetic warm-up mechanism addressing the sensitivity of F2M's neural field to initial poses.
- For **Reviewer Qbae**, we replaced the term "bootstrap" with "warm-up" throughout the paper to better describe the pre-training process. This modification is more pertinent to the claim that pre-training on a synthetic dataset provide a better initialization for F2M approach to achieve better performance.
- For **Reviewer bx3G**, we added a comparison table between F2M-Reg and PointMBF on TUM RGB-D in the Appendix. Additionally, we included a **Limitation** section in Section 6 to acknowledge and discuss potential weaknesses in our method.

We will continue to address the reviewers' concerns comprehensively and provide detailed responses.

---

### Author Response · Authors · 2024-12-04
**Summary of the Disccusion**

Dear Chairs and Reviewers,

Hope this message finds you well.

As the discussion period comes to a close, we would like to provide a brief summary of our discussions with the reviewers for reference. First and foremost, we sincerely thank all the reviewers for their insightful comments and valuable suggestions.

We summarize the main concerns raised by the reviewers along with our corresponding responses as follows:

- **About the performance between F2M-Reg w/o warm up and PointMBF**.  As detailed in the main paper, the frame-to-model approach is highly sensitive to model initialization, which we address with the synthetic warm-up mechanism. Additional results from various training datasets (ScanNet, 3DMatch) and evaluation settings (20/50-frame apart) further demonstrate the superiority of our method.
- **About the writing.** We offer a deeper explanation of the pipeline and its individual modules, including further discussion on the differences between zero-shot learning and unsupervised registration in our method.  We also clarify and modify ambiguous terms and provide a more detailed description of the advantages of the frame-to-model approach over the frame-to-frame approach, supported by extensive experimental evidence.
- **About the usage of Co-SLAM like approach in the fine-tuning stage.** This issue extends to the question of whether training datasets need to be sequential and the scope of our method. Notably, our approach can train the registration model under low-overlap conditions, where the assumption of constant movement no longer holds. The registration model then provides the initial pose necessary for constructing the neural implicit field.
- **About more ablation studies.**  We attribute the improvement to both warming up and fine-tuning stages. To identify the specific contributions of each, we conduct ablation studies on warming-up dataset size, object count per scene, model generalizability, and rendering strategies. Furthermore, we evaluated our method and PointMBF under the 20-frame-apart and 50-frame-apart settings.

---

Based on the discussion with reviewers, we also present a brief summary of our paper  as follows:

- **Observation:** Existing frame-to-frame methods usually adopt a pairwise training strategy based on differentiable rendering, which suffers from poor multi-view consistency due to factors such as lighting changes, geometry occlusion and reflective materials.
- **Solution:** F2M-Reg addresses these issues by introducing a frame-to-model optimization framework, which leverages the neural implicit filed as a global model of the scene and use the consistency between the input and the re-rendered frames for pose optimization. To facilitate the neural field optimization, F2M-Reg warms up the registration model on synthetic dataset, Sim_RGBD.
- **Results:** F2M-Reg outperforms the state-of-the-art counterparts on two popular indoor RGB-D datasets, ScanNet and 3DMatch. Also F2M-Reg achieves significantly better performance than previous methods in more challenging scenarios with lower overlap or severe lighting changes.
- **Highlights:**  Focusing on unsupervised RGB-D registration task, our work has the following highlights:
  - **Frame-to-model optimization framework:** The infusion of global reconstruction information enhances the reliability of re-rendering, which fortifies the robustness of registration model.
  - **Synthetic warm-up mechanism:** The warming up mechanism  provide high-quality initial poses for neural implicit field optimization.

Thank you once again for your efforts in reviewing and discussing our work. We greatly appreciate all the valuable feedback that contributed to enhancing our submission.

Sincerely
Authors of Submission 7404

---

### Meta-Review · Area_Chair_6wN4 · 2024-12-21

**Metareview:**

The paper proposes an unsupervised method for robust RGB-D registration without ground-truth pose supervision where a frame-to-model approach is employed. The proposed method begins with pre-training the model on a synthetic dataset, Sim-RGBD, with ground-truth poses, and subsequently fine-tunes it on real-world datasets without ground-truth poses by leveraging a neural implicit field as a 3D scene representation for pose optimization.

**Additional Comments On Reviewer Discussion:**

The usage of a neural implicit field as a global scene model to capture broader scene-level information is nice to handle complex conditions, such as low overlap. The proposed method demonstrates high performance compared to baseline methods on two real-world RGB-D datasets. On the other hand, the reviewers raised concerns regarding insufficient validation of the proposed method, lacking analysis of experimental results, undetailed explanations about the proposed method.  The authors have provided additional experiments and their analysis to address the raised concerns.  Further discussion reveals the improvement mostly comes from the extra training data and improvement is significant for 50-frame apart setting while 20-frame apart case is not.  In the end, the final scores of the reviewers are splitting: 2 x BR, A, and BA. The remaining concerns by the negative reviewers are still insufficient validation and technical novelty and depth.  Evaluations on ScanNet and 3DMatch are not thorough. As the authors have admitted, the performance gain from Sim-RGBD warm-up across two datasets is inconsistent. The authors reason that (1) domain gap because ScanNet is used for testing in all the experiment, and (2) difference of dataset size.  The domain gap can be easily evidenced by swapping the roles of ScanNet and 3DMatch in the experiments, which is missing.  New experiment on TUM RGB is only for 50-frame setting while that on ScanNet++ is only for 20-frame setting.  Why not both settings on both datasets?  With further results, the generalization ability of the proposed method under high/low overlap across datasets could have been discussed.  Applicability to unordered data (Reviewer bx3G’s comment) is not evidenced.  The influence of the dataset size for worming-up is newly provided; however, the performance behaviors are not aligned with the size (up and down).  Nevertheless, the authors observe that increasing the dataset size increase the performance although improvement is not significant. Regarding the comparison between F2M and F2F with warm-up, the performance difference under 50-frame setting significant while that under 20-frame setting is negligible.  In-depth discussion on this observation is not addressed.  As seen, the validation of the proposed method is not thorough yet.  Reviewer 4zeD acknowledges the effectiveness in correspondence matching and pose estimation under large viewpoint change (50 frames), though under common viewpoint change setting in existing work (20 frames), the performance is on par with SOTA, and still has major concern on the technical novelty and depth. Reviewer 4zeD commented as below in the AC-reviewer discussion phase:

1.	The main contribution of the work is leveraging a scene and pose optimization method (like SLAM, SFM with NeRF) to provide the pose label for the correspondence learning of RGBD image pairs. This strategy is one of main solutions to acquire GT for correspondence learning, for example using COLMAP to get pose and matching points for image pairs. This work replaces the COLMAP with more recent NeRF based SfM method. Compared with SOTA methods e.g. PointMBF which only use mutual RGB or depth consistency between image pairs without knowledge about the scene in COLMAP or NeRF based SFM, these GT for SfM is actually oracle.

2.	The other concern is its claim of the contribution of the synthetic data. I acknowledge that using synthetic data will definitely help the optimization of NeRF based SfM, but I cannot see in which sense it fills the gap of the existing synthetic or real dataset. Pretraining in synthetic data with randomly place objects for point matching is also quite common [1].
[1] 2018 ECCV RAFT: Recurrent All-Pairs Field Transforms for Optical Flow

AC shares with Reviewer4zeD’s observation that the meaningful improvement is only for 50-frame setting (low overlap).  Then, performance gain under different low overlap scenarios should be systematically evaluated to confirm this observation further, leading to more proper claim rather than that the proposed method overcomes SOTAs.  More in-depth arguments to clarify the real contributions of this work should be addressed to convince the reviewers. AC also finds that discussion between the authors and the reviewers has not been adequately reflected in the revised manuscript, meaning the final version will be largely different from the current version, and another round of reviews will be required. On balance, AC finds the remaining concerns to outweigh the current technical contributions, and agrees that the paper would benefit from more work.  For this, the paper cannot be accepted to this conference.

---

### Decision · Program_Chairs · 2025-01-22

Reject